# ADAPTIVE GRADIENT METHODS CONVERGE FASTER WITH OVER-PARAMETERIZATION (AND YOU CAN DO A LINE-SEARCH)

## ABSTRACT

Adaptive gradient methods are typically used for training over-parameterized models capable of exactly fitting the data; we thus study their convergence in this *interpolation* setting. Under an interpolation assumption, we prove that AMSGrad with a constant step-size and momentum can converge to the minimizer at the faster $\mathcal{O}(1/T)$ rate for smooth, convex functions. Furthermore, in this setting, we show that AdaGrad can achieve an $\mathcal{O}(1)$ regret in the online convex optimization framework. When interpolation is only approximately satisfied, we show that constant step-size AMSGrad converges to a neighbourhood of the solution. On the other hand, we prove that AdaGrad is robust to the violation of interpolation and converges to the minimizer at the optimal rate. However, we demonstrate that even for simple, convex problems satisfying interpolation, the empirical performance of these methods heavily depends on the step-size and requires tuning. We alleviate this problem by using stochastic line-search (SLS) and Polyak's step-sizes (SPS) to help these methods adapt to the function's local smoothness. By using these techniques, we prove that AdaGrad and AMSGrad do not require knowledge of problem-dependent constants and retain the convergence guarantees of their constant step-size counterparts. Experimentally, we show that these techniques help improve the convergence and generalization performance across tasks, from binary classification with kernel mappings to classification with deep neural networks.

## 1 INTRODUCTION

Adaptive gradient methods such as AdaGrad (Duchi et al., 2011), RMSProp (Tieleman & Hinton, 2012), AdaDelta (Zeiler, 2012), Adam (Kingma & Ba, 2015), and AMSGrad (Reddi et al., 2018) are popular optimizers for training deep neural networks (Goodfellow et al., 2016). These methods scale well and exhibit good performance across problems, making them the default choice for many machine learning applications. Theoretically, these methods are usually studied in the non-smooth, online convex optimization setting (Duchi et al., 2011; Reddi et al., 2018) with recent extensions to the strongly-convex (Mukkamala & Hein, 2017; Wang et al., 2020; Xie et al., 2020) and non-convex settings (Li & Orabona, 2019; Ward et al., 2019; Zhou et al., 2018; Chen et al., 2019; Wu et al., 2019; Défossez et al., 2020; Staib et al., 2019). An online–batch reduction gives guarantees similar to stochastic gradient descent (SGD) in the offline setting (Cesa-Bianchi et al., 2004; Hazan & Kale, 2014; Levy et al., 2018).

However, there are several discrepancies between the theory and application of these methods. Although the theory advocates for using decreasing step-sizes for Adam, AMSGrad and its variants (Kingma & Ba, 2015; Reddi et al., 2018), a constant step-size is typically used in practice (Paszke et al., 2019). Similarly, the standard analysis of these methods assumes a decreasing momentum parameter, however, the momentum is fixed in practice. On the other hand, AdaGrad (Duchi et al., 2011) has been shown to be "universal" as it attains the best known convergence rates in both the stochastic smooth and non-smooth settings (Levy et al., 2018), but its empirical performance is rather disappointing when training deep models (Kingma & Ba, 2015). Improving the empirical performance was indeed the main motivation behind Adam and other methods (Tieleman & Hinton, 2012; Zeiler, 2012) that followed AdaGrad. Although these methods have better empirical performance, they are not guaranteed to converge to the solution with a constant step-size and momentum parameter.

Another inconsistency is that although the standard theoretical results are for non-smooth functions, these methods are also extensively used in the easier, smooth setting. More importantly, adaptive gradient methods are generally used to train highly expressive, large over-parameterized models (Zhang et al., 2017; Liang & Rakhlin, 2018) capable of interpolating the data. However, the standard theoretical analyses do not take advantage of these additional properties. On the other hand, a line of recent work (Schmidt & Le Roux, 2013; Jain et al., 2018; Ma et al., 2018; Liu & Belkin, 2020; Cevher & Vũ, 2019; Vaswani et al., 2019a;b; Wu et al., 2019; Loizou et al., 2020) focuses on the convergence of SGD in this *interpolation* setting. In the standard finite-sum case, interpolation implies that all the functions in the sum are minimized at the same solution. Under this additional assumption, these works show SGD with a constant step-size converges to the minimizer at a faster rate for both convex and non-convex smooth functions.

In this work, we aim to resolve some of the discrepancies in the theory and practice of adaptive gradient methods. To theoretically analyze these methods, we consider a simplistic setting - smooth, convex functions under interpolation. Using the intuition gained from theory, we propose better techniques to adaptively set the step-size for these methods, dramatically improving their empirical performance when training over-parameterized models.

## 1.1 BACKGROUND AND CONTRIBUTIONS

**Constant step-size.** We focus on the theoretical convergence of two adaptive gradient methods: AdaGrad and AMSGrad. For smooth, convex functions, Levy et al. (2018) prove that AdaGrad with a constant step-size adapts to the smoothness and gradient noise, resulting in an $\mathcal{O}(1/T + \varsigma/\sqrt{T})$ convergence rate, where $T$ is the number of iterations and $\varsigma^2$ is a global bound on the variance in the stochastic gradients. This convergence rate matches that of SGD under the same setting (Moulines & Bach, 2011). In Section 3, we show that constant step-size AdaGrad also adapts to interpolation and prove an $\mathcal{O}(1/T + \sigma/\sqrt{T})$ rate, where $\sigma$ is the extent to which interpolation is violated. In the over-parameterized setting, $\sigma^2$ can be much smaller than $\varsigma^2$ (Zhang & Zhou, 2019), implying a faster convergence. When interpolation is exactly satisfied, $\sigma^2 = 0$, we obtain an $\mathcal{O}(1/T)$ rate, while $\varsigma^2$ can still be large. In the online convex optimization framework, for smooth functions, we show that the regret of AdaGrad improves from $\mathcal{O}(\sqrt{T})$ to $\mathcal{O}(1)$ when interpolation is satisfied and retains its $\mathcal{O}(\sqrt{T})$-regret guarantee in the general setting (Appendix C.2). Assuming its corresponding preconditioner remains bounded, we show that AMSGrad with a constant step-size and constant momentum parameter also converges at the rate $\mathcal{O}(1/T)$ under interpolation (Section 4). However, unlike AdaGrad, it requires specific step-sizes that depend on the problem's smoothness. More generally, constant step-size AMSGrad converges to a neighbourhood of the solution, attaining an $\mathcal{O}(1/T + \sigma^2)$ rate, which matches the rate of constant step-size SGD in the same setting (Schmidt & Le Roux, 2013; Vaswani et al., 2019a). When training over-parameterized models, this result provides some justification for the faster ($\mathcal{O}(1/T)$ vs. $\mathcal{O}(1/\sqrt{T})$) convergence of the AMSGrad variant typically used in practice.

**Adaptive step-size.** Although AdaGrad converges at the same asymptotic rate for any step-size (up to constants), it is unclear how to choose this step-size without manually trying different values. Similarly, AMSGrad is sensitive to the step-size, converging only for a specific range in both theory and practice. In Section 5, we experimentally show that even for simple, convex problems, the step-size has a big impact on the empirical performance of AdaGrad and AMSGrad. To overcome this limitation, we use recent methods (Vaswani et al., 2019a; Loizou et al., 2020) that automatically set the step-size for SGD. These works use stochastic variants of the classical Armijo line-search (Armijo, 1966) or the Polyak step-size (Polyak, 1963) in the interpolation setting. We combine these techniques with adaptive gradient methods and show that a variant of stochastic line-search (SLS) enables AdaGrad to adapt to the smoothness of the underlying function, resulting in faster empirical convergence, while retaining its favourable convergence properties (Section 3). Similarly, AMSGrad with variants of SLS and SPS can match the convergence rate of its constant step-size counterpart, but without knowledge of the underlying smoothness properties (Section 4).

**Experimental results.** Finally, in Section 5, we benchmark our results against SGD variants with SLS (Vaswani et al., 2019b), SPS (Loizou et al., 2020), tuned Adam and its recently proposed variants (Luo et al., 2019; Liu et al., 2020). We demonstrate that the proposed techniques for setting the step-size improve the empirical performance of adaptive gradient methods. These improvements are consistent across tasks, ranging from binary classification with a kernel mapping to multi-class classification using standard deep neural network architectures.

## 2 PROBLEM SETUP

We consider the unconstrained minimization of an objective $f : \mathbb{R}^d \to \mathbb{R}$ with a finite-sum structure, $f(w) = \frac{1}{n}\sum_{i=1}^{n} f_i(w)$. In supervised learning, $n$ represents the number of training examples, and $f_i$ is the loss function on training example $i$. Although we focus on the finite-sum setting, our results can be easily generalized to the online optimization setting. The objective of our analysis is to better understand the effect of the step-size and line-searches when interpolation is (almost) satisfied. This is complicated by the fact that adaptive methods are still poorly understood; state-of-the-art analyses do not show an improvement over gradient descent in the worst-case. To focus on the effect of step-sizes, we make the simplifying assumptions described in this section.

We assume $f$ and each $f_i$ are differentiable, convex, and lower-bounded by $f^*$ and $f_i^*$, respectively. Furthermore, we assume that each function $f_i$ in the finite-sum is $L_i$-smooth, implying that $f$ is $L_{\max}$-smooth, where $L_{\max} = \max_i L_i$. We also make the standard assumption that the iterates remain bounded in a ball of radius $D$ around a global minimizer, $\|w_k - w^*\| \le D$ for all $w_k$ (Ahn et al., 2020). We remark that the bounded iterates assumption simplifies the analysis but is not essential, and similar to Reddi et al. (2018); Duchi et al. (2011); Levy et al. (2018), our theoretical results can be extended to include a projection step. We include the formal definitions of these properties (Nemirovski et al., 2009) in Appendix A.

The interpolation assumption means that the gradient of *each* $f_i$ in the finite-sum converges to zero at an optimum. If the overall objective $f$ is minimized at $w^*$, $\nabla f(w^*) = 0$, then for all $f_i$ we have $\nabla f_i(w^*) = 0$. The interpolation condition can be exactly satisfied for many over-parameterized machine learning models such as non-parametric kernel regression without regularization (Belkin et al., 2019; Liang & Rakhlin, 2018) and over-parameterized deep neural networks (Zhang et al., 2017). We measure the extent to which interpolation is violated by the disagreement between the minimum overall function value $f^*$ and the minimum value of each individual functions $f_i^*$, $\sigma^2 := \mathbb{E}_i[f^* - f_i^*] \in [0, \infty)$ (Loizou et al., 2020). The minimizer of $f$ need not be unique for $\sigma^2$ to be uniquely defined, as it only depends on the minimum function values. Interpolation is said to be exactly satisfied if $\sigma^2 = 0$, and we also study the setting when $\sigma^2 > 0$.

For a preconditioner matrix $A_k$ and a constant momentum parameter $\beta \in [0, 1)$, the update for a generic adaptive gradient method at iteration $k$ can be expressed as:

$$w_{k+1} = w_k - \eta_k A_k^{-1} m_k \quad ; \quad m_k = \beta m_{k-1} + (1-\beta)\nabla f_{i_k}(w_k) \tag{1}$$

Here, $\nabla f_{i_k}(w_k)$ is the stochastic gradient of a randomly chosen function $f_{i_k}$, and $\eta_k$ is the step-size. Adaptive gradient methods typically differ in how their preconditioners are constructed and whether or not they include the momentum term $\beta m_{k-1}$ (see Table 1 for a list of common methods). Both

Table 1: Adaptive preconditioners (analyzed methods are **bolded**), with $G_0 = 0$ and $\beta_1, \beta_2 \in [0, 1)$. In practice, a small $\epsilon I$ is added to ensure $A_k \succ 0$. *: We use the PyTorch implementation in experiments which includes bias correction.

| Optimizer | $G_k$ $\quad(\nabla_k := \nabla f_{i_k}(w_k))$ | | $A_k$ | $\beta$ |
|---|---|---|---|---|
| **AdaGrad** | $G_{k-1} + \mathrm{diag}(\nabla_k \nabla_k^\top)$ | | $G_k^{1/2}$ | 0 |
| RMSProp | $\beta_2 G_{k-1} + (1-\beta_2)\,\mathrm{diag}(\nabla_k \nabla_k^\top)$ | | $G_k^{1/2}$ | 0 |
| Adam | $(\beta_2 G_{k-1} + (1-\beta_2)\,\mathrm{diag}(\nabla_k \nabla_k^\top))/(1-\beta_2^k)$ | | $G_k^{1/2}$ | $\beta_1$ |
| **AMSGrad*** | $(\beta_2 G_{k-1} + (1-\beta_2)\,\mathrm{diag}(\nabla_k \nabla_k^\top))/(1-\beta_2^k)$ | | $\max\{A_{k-1}, G_k^{1/2}\}$ | $\beta_1$ |

RMSProp and Adam maintain an exponential moving average of past stochastic gradients, but as Reddi et al. (2018) pointed out, unlike AdaGrad, the corresponding preconditioners do not guarantee that $A_{k+1} \succeq A_k$ and the resulting per-dimension step-sizes do not go to zero. This can lead to large fluctuations in the effective step-size and prevent these methods from converging. To mitigate this problem, they proposed AMSGrad, which ensures $A_{k+1} \succeq A_k$ and the convergence of iterates. Consequently, our theoretical results focus on AdaGrad, AMSGrad and other adaptive gradient methods that ensure this monotonicity. However, we also considered RMSProp and Adam in our experimental evaluation.

Although our theory holds for both the full matrix and diagonal variants (where $A_k$ is a diagonal matrix) of these methods, we use only the latter in experiments for scalability. The diagonal variants

perform a per-dimension scaling of the gradient and avoid computing the full matrix inverse, so their per-iteration cost is the same as SGD, although with an additional $\mathcal{O}(d)$ memory. For AMSGrad, we assume that the corresponding preconditioners are well-behaved in the sense that their eigenvalues are bounded in an interval $[a_{\min}, a_{\max}]$. This is a common assumption made in the analysis of adaptive methods. Moreover, for diagonal preconditioners, such a boundedness property is easy to verify, and it is also inexpensive to maintain the desired range by projection. Our main theoretical results for AdaGrad (Section 3) and AMSGrad (Section 4) are summarized in Table 2.

Table 2: Results for smooth, convex functions.

| Method | Step-size | Adapts to smoothness | Rate | Reference |
|---|---|:---:|---|---|
| AdaGrad | Constant | ✗ | $\mathcal{O}(1/T + \sigma/\sqrt{T})$ | Theorem 1 |
| | Conservative Lipschitz LS | ✓ | $\mathcal{O}(1/T + \sigma/\sqrt{T})$ | Theorem 2 |
| AMSGrad | Constant | ✗ | $\mathcal{O}(1/T + \sigma^2)$ | Theorem 3 |
| AMSGrad w/o momentum | Armijo SLS | ✓ | $\mathcal{O}(1/T + \sigma^2)$ | Theorem 4 |
| AMSGrad | Conservative Armijo SPS | ✓ | $\mathcal{O}(1/T + \sigma^2)$ | Theorem 5 |

## 3 ADAGRAD

For smooth, convex objectives, Levy et al. (2018) showed that AdaGrad converges at a rate $\mathcal{O}(1/T + \zeta/\sqrt{T})$, where $\zeta^2 = \sup_w \mathbb{E}_i[\|\nabla f(w) - \nabla f_i(w)\|^2]$ is a uniform bound on the variance of the stochastic gradients. In the over-parameterized setting, we show that AdaGrad achieves the $\mathcal{O}(1/T)$ rate when interpolation is exactly satisfied and a slower convergence to the solution if interpolation is violated.[1] The proofs for this section are in Appendix C.

**Theorem 1** (Constant step-size AdaGrad). *Assuming (i) convexity and (ii) $L_{\max}$-smoothness of each $f_i$, and (iii) bounded iterates, AdaGrad with a constant step-size $\eta$ and uniform averaging such that $\bar{w}_T = \frac{1}{T} \sum_{k=1}^{T} w_k$, converges at a rate*

$$\mathbb{E}[f(\bar{w}_T) - f^*] \leq \frac{\alpha}{T} + \frac{\sqrt{\alpha}\sigma}{\sqrt{T}}, \quad \text{where } \alpha = \frac{1}{2}\left(\frac{D^2}{\eta} + 2\eta\right)^2 dL_{\max}.$$

When interpolation is exactly satisfied, a similar proof technique can be used to show that AdaGrad incurs only $O(1)$ regret in the online convex optimization setting (Theorem 6 in Appendix C.2). The above theorem shows that AdaGrad is robust to the violation of interpolation and converges to the minimizer at the desired rate for *any* reasonable step-size. Although this is a favourable property, the best constant step-size depends on the problem, and as we demonstrate experimentally in Section 5, the performance of AdaGrad depends on correctly tuning this step-size.

To overcome this limitation, we use a *conservative Lipschitz line-search* that sets the step-size on the fly, improving the empirical performance of AdaGrad while retaining its favourable convergence guarantees. At each iteration, this line-search selects a step-size $\eta_k$ that satisfies the property

$$f_{i_k}(w_k - \eta_k \nabla f_{i_k}(w_k)) \leq f_{i_k}(w_k) - c\,\eta_k \|\nabla f_{i_k}(w_k)\|^2, \quad \text{and } \eta_k \leq \eta_{k-1}. \tag{2}$$

The resulting step-size is then used in the standard AdaGrad update in Eq. (1). To find an acceptable step, our results use a backtracking line-search, described in Appendix F. For simplicity, the theoretical results assume access to the largest step-size that satisfies the above condition.[2] Here, $c$ is a hyper-parameter determined theoretically and typically set to $1/2$ in our experiments. The "conservative" part of the line-search is the non-increasing constraint on the step-sizes, which is essential for convergence to the minimizer when interpolation is violated. We refer to it as the *Lipschitz line-search* as it is only used to estimate the local Lipschitz constant. Unlike the classical Armijo line-search for

---

[1] A similar result also appears in the course notes (Orabona, 2019).
[2] The difference between the exact and backtracking line-search is minimal, and the bounds are only changed by a constant depending on the backtracking parameter.

preconditioned gradient descent, the line-search in Eq. (2) is in the gradient direction, even though the update is in the preconditioned direction. The resulting step-size found is guaranteed to be in the range $[2(1-c)/L_{\max}, \eta_{k-1}]$ (Vaswani et al., 2019b) and allows us to prove the following theorem.

**Theorem 2.** *Under the same assumptions as Theorem 1, AdaGrad with a conservative Lipschitz line-search with $c = 1/2$, a step-size upper bound $\eta_{\max}$ and uniform averaging converges at a rate*

$$\mathbb{E}[f(\bar{w}_T) - f^*] \leq \frac{\alpha}{T} + \frac{\sqrt{\alpha}\sigma}{\sqrt{T}}, \quad where \ \alpha = \frac{1}{2}\left(D^2 \max\left\{\frac{1}{\eta_{\max}}, L_{\max}\right\} + 2\,\eta_{\max}\right)^2 dL_{\max}.$$

Intuitively, the Lipschitz line-search enables AdaGrad to take larger steps at iterates where the underlying function is smoother. It retains the favourable convergence guarantees of constant step-size AdaGrad, while improving its empirical performance (Section 5). Moreover, if interpolation is exactly satisfied, we can obtain an $\mathcal{O}(1/T)$ convergence without the conservative constraint $\eta_k \leq \eta_{k-1}$ on the step-sizes (Appendix C.3).

# 4 AMSGRAD AND NON-DECREASING PRECONDITIONERS

In this section, we consider AMSGrad and, more generally, methods with non-decreasing preconditioners satisfying $A_k \succeq A_{k-1}$. As our focus is on the behavior of the algorithm with respect to the overall step-size, we make the simplifying assumption that the effect of the preconditioning is bounded, meaning that the eigenvalues of $A_k$ lie in the $[a_{\min}, a_{\max}]$ range. This is a common assumption made in the analyses of adaptive methods (Reddi et al., 2018; Alacaoglu et al., 2020) that prove worst-case convergence rates matching those of SGD. For our theoretical results, we consider the variant of AMSGrad without bias correction, as its effect is minimal after the first few iterations. The proofs for this section are in Appendix D and Appendix E.

The original analysis of AMSGrad (Reddi et al., 2018) uses a decreasing step-size and a decreasing momentum parameter. It shows an $\mathcal{O}(1/\sqrt{T})$ convergence for AMSGrad in both the smooth and non-smooth convex settings. Recently, Alacaoglu et al. (2020) showed that this analysis is loose and that AMSGrad does not require a decreasing momentum parameter to obtain the $\mathcal{O}(1/\sqrt{T})$ rate. However, in practice, AMSGrad is typically used with both a constant step-size and momentum parameter. Next, we present the convergence result for this commonly-used variant of AMSGrad.

**Theorem 3.** *Under the same assumptions as Theorem 1, and assuming (iv) non-decreasing preconditioners (v) bounded eigenvalues in the $[a_{\min}, a_{\max}]$ interval, where $\kappa = a_{\max}/a_{\min}$, AMSGrad with $\beta \in [0, 1)$, constant step-size $\eta = \frac{1-\beta}{1+\beta}\frac{a_{\min}}{2L_{\max}}$ and uniform averaging converges at a rate,*

$$\mathbb{E}[f(\bar{w}_T) - f^*] \leq \left(\frac{1+\beta}{1-\beta}\right)^2 \frac{2L_{\max}D^2 d\kappa}{T} + \sigma^2.$$

When $\sigma = 0$, we obtain a $\mathcal{O}(1/T)$ convergence to the minimizer. However, when interpolation is only approximately satisfied, we obtain convergence to a neighbourhood with its size depending on $\sigma^2$. We observe that the noise $\sigma^2$ is not amplified because of the non-decreasing momentum (or step-size). A similar distinction between the convergence of constant step-size Adam (or AMSGrad) vs. AdaGrad has also been recently discussed in the non-convex setting (Défossez et al., 2020). Unfortunately, the final bound is minimized by setting $\beta_1 = 0$ and our theoretical analysis does not show an advantage of using momentum. Note that this is a common drawback in the analyses of heavy-ball momentum for non-quadratic functions in both the stochastic and deterministic settings (Ghadimi et al., 2015; Reddi et al., 2018; Alacaoglu et al., 2020; Sebbouh et al., 2020).

Since AMSGrad is typically used for optimizing over-parameterized models, the violation $\sigma^2$ is small, even when interpolation is not exactly satisfied. Another reason that constant step-size AMSGrad is practically useful is because of the use of large batch-sizes that result in a smaller effective neighbourhood. To get some intuition about the effect of batch-size, note that if we use a batch-size of $b$, the resulting neighbourhood depends on $\sigma_b^2 := \mathbb{E}_{B;|B|=b}[f_B(w^*) - f_B(x_B^*)]$ where $w_B^*$ is the minimizer of a batch $B$ of training examples. By convexity, $\sigma_b^2 \leq \mathbb{E}[\|w^* - x_B^*\|\,\|\nabla f_B(w^*)\|]$. If we assume that the distance $\|w^* - x_B^*\|$ is bounded, $\sigma_b^2 \propto \mathbb{E}\|\nabla f_B(w^*)\|$. Since the examples in each batch are sampled with replacement, using the bounds in (Lohr, 2009), $\sigma_b^2 \propto \frac{n-b}{nb}\|\nabla f_i(w^*)\|$, showing that the effective neighbourhood shrinks as the batch-size becomes larger, becoming zero for

the full-batch variant. With over-parameterization and large batch-sizes, the effective neighbourhood is small enough for machine learning tasks that do not require exact convergence to the solution.

The constant step-size required for the above result depends on $L_{\max}$, which is typically unknown. Furthermore, using a global bound on $L_{\max}$ usually results in slower convergence since the local Lipschitz constant can vary considerably during the optimization. To overcome these issues, we use a stochastic variant of the *Armijo line-search*. Unlike the Lipschitz line-search whose sole purpose is to estimate the Lipschitz constant, the Armijo line-search selects a suitable step-size in the preconditioned gradient direction, and as we show in Section 5, it results in better empirical performance. Similar to the constant step-size, when interpolation is violated, we only obtain convergence to a neighbourhood of the solution. The stochastic Armijo line-search returns the largest step-size $\eta_k$ satisfying the following conditions at iteration $k$,

$$f_{i_k}(w_k - \eta_k A_k^{-1}\nabla f_{i_k}(w_k)) \leq f_{i_k}(w_k) - c\,\eta_k \left\|\nabla f_{i_k}(w_k)\right\|_{A_k^{-1}}^2, \quad \text{and } \eta_k \leq \eta_{\max}. \quad (3)$$

The step-size is artificially upper-bounded by $\eta_{\max}$ (typically chosen to be a large value). The line-search guarantees descent on the current function $f_{i_k}$ and that $\eta_k$ lies in the $\left[2a_{\min}(1-c)/L_{\max}, \eta_{\max}\right]$ range. In the next theorem, we first consider the variant of AMSGrad without momentum ($\beta = 0$) and show that using the Armijo line-search retains the $O(1/T)$ convergence rate without the need to know the Lipschitz constant.

**Theorem 4.** *Under the same assumptions as Theorem 1, AMSGrad with zero momentum, Armijo line-search with $c = 3/4$, a step-size upper bound $\eta_{\max}$ and uniform averaging converges at a rate,*

$$\mathbb{E}[f(\bar{w}_T) - f^*] \leq \left(\frac{3D^2 d \cdot a_{\max}}{2T} + 3\eta_{\max}\sigma^2\right)\max\left\{\frac{1}{\eta_{\max}}, \frac{2L_{\max}}{a_{\min}}\right\}.$$

Comparing this rate with that of using constant step-size (Theorem 3), we observe that the Armijo line-search results in a worse constant in the convergence rate and a larger neighbourhood. These dependencies can be improved by considering a conservative version of the Armijo line-search. However, we experimentally show that the proposed line-search drastically improves the empirical performance of AMSGrad. We show that a similar bound also holds for AdaGrad (see Theorem 7 in Appendix C). AdaGrad with an Armijo line-search converges to a neighbourhood in the absence of interpolation (unlike the results in 3). Moreover, the above bound depends on $a_{\min}$ which can be $\mathcal{O}(\epsilon)$ in the worst-case, resulting in an unsatisfactory worst-case rate of $\mathcal{O}(1/\epsilon T)$ even in the interpolation setting. However, like AMSGrad, AdaGrad with Armijo line-search has excellent empirical performance, implying the need for a different theoretical assumption in the future.

Before considering techniques to set the step-size for AMSGrad including momentum, we present the details of the stochastic Polyak step-size (SPS) Loizou et al. (2020); Berrada et al. (2019) and Armijo SPS, our modification to the adaptive setting. These variants set the step-size as:

$$\text{SPS}: \eta_k = \min\left\{\frac{f_{i_k}(w_k) - f_{i_k}^*}{c\left\|\nabla f_{i_k}(w_k)\right\|^2}, \eta_{\max}\right\}, \quad \text{Armijo SPS}: \eta_k = \min\left\{\frac{f_{i_k}(w_k) - f_{i_k}^*}{c\left\|\nabla f_{i_k}(w_k)\right\|_{A_k^{-1}}^2}, \eta_{\max}\right\}.$$

Here, $f_{i_k}^*$ is the minimum value for the function $f_{i_k}$. The advantage of SPS over a line-search is that it does not require a potentially expensive backtracking procedure to set the step-size. Moreover, it can be shown that this step-size is always larger than the one returned by line-search, which can lead to faster convergence. However, SPS requires knowledge of $f_i^*$ for each function in the finite-sum. This value is difficult to obtain for general functions but is readily available in the interpolation setting for many machine learning applications. Common loss functions are lower-bounded by zero, and the interpolation setting ensures that these lower-bounds are tight. Consequently, using SPS with $f_i^* = 0$ has been shown to yield good performance for over-parameterized problems (Loizou et al., 2020; Berrada et al., 2019). In Appendix D, we show that the Armijo line-search used for the previous results can be replaced by Armijo SPS and result in similar convergence rates.

For AMSGrad with momentum, we propose to use a *conservative* variant of Armijo SPS that sets $\eta_{\max} = \eta_{k-1}$ at iteration $k$ ensuring that $\eta_k \leq \eta_{k-1}$. This is because using a potentially increasing step-size sequence along with momentum can make the optimization unstable and result in divergence. Using this step-size, we prove the following result.

**Theorem 5.** *Under the same assumptions of Theorem 1 and assuming (iv) non-decreasing preconditioners (v) bounded eigenvalues in the $[a_{\min}, a_{\max}]$ interval with $\kappa = a_{\max}/a_{\min}$, AMSGrad with*

$\beta \in [0, 1)$*, conservative Armijo SPS with $c = {}^{1+\beta}/{1-\beta}$ and uniform averaging converges at a rate,*

$$\mathbb{E}[f(\bar{w}_T) - f^*] \leq \left(\frac{1+\beta}{1-\beta}\right)^2 \frac{2L_{\max}D^2 d\kappa}{T} + \sigma^2.$$

The above result exactly matches the convergence rate in Theorem 3 but does not require knowledge of the smoothness constant to set the step-size. Moreover, the conservative step-size enables convergence without requiring an artificial upper-bound $\eta_{\max}$ as in Theorem 8. We note that a similar convergence rate can be obtained when using a conservative variant of Armijo SLS ( Appendix E.2), although our theoretical techniques only allow for a restricted range of $\beta$.

When $A_k = I_d$, the AMSGrad update is equivalent to the update for SGD with heavy-ball momentum (Sebbouh et al., 2020). By setting $A_k = I_d$ in the above result, we recover an $O(1/T + \sigma^2)$ rate for SGD (using SPS to set the step-size) with heavy-ball momentum. In the smooth, convex setting, our rate matches that of (Sebbouh et al., 2020); however, unlike their result, we do not require knowledge of the Lipschitz constant. This result also provides theoretical justification for the heuristic used for incorporating heavy-ball momentum for SLS in (Vaswani et al., 2019b).

For a general preconditioner, the AMSGrad update in Eq. (1) is not equivalent to heavy-ball momentum. With a constant momentum parameter $\gamma \in [0, 1)$, the general heavy-ball update (Loizou & Richtárik, 2017) is given as $w_{k+1} = w_k - \alpha_k A_k^{-1} \nabla f_{i_k}(w_k) + \gamma (w_k - w_{k-1})$ (refer to Appendix E.1 for a relation between the two updates). Unlike this update, AMSGrad also preconditions the momentum direction $(w_k - w_{k-1})$. If we consider the zero-momentum variant of adaptive gradient methods as preconditioned gradient descent, the above update is a more natural way to incorporate momentum. We explore this alternate method and prove the same $O(1/T + \sigma^2)$ convergence rate for constant step-size, conservative Armijo SPS and Armijo SLS techniques in Appendix E.3. In the next section, we use the above techniques for training large over-parameterized deep networks.

## 5 EXPERIMENTAL EVALUATION

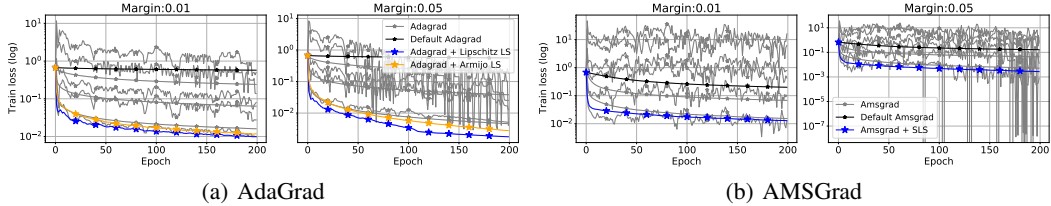

(a) AdaGrad        (b) AMSGrad

Figure 1: Synthetic experiments showing the impact of step-size on the performance of AdaGrad, AMSGrad with varying step-sizes, including the default in PyTorch, and the SLS variants.

**Synthetic experiment:** We first present an experiment to show that AdaGrad and AMSGrad with constant step-size are not robust even for simple, convex problems. We use their PyTorch implementations (Paszke et al., 2019) on a binary classification task with logistic regression. Following the protocol of Meng et al. (2020), we generate a linearly-separable dataset with $n = 10^3$ examples (ensuring interpolation is satisfied) and $d = 20$ features with varying margins. For AdaGrad and AMSGrad with a batch-size of 100, we show the training loss for a grid of step-sizes in the $[10^3, 10^{-3}]$ range and also plot their default (in PyTorch) variants. For AdaGrad, we compare against the proposed Lipschitz line-search and Armijo SLS variants. As is suggested by the theory, for each of these variants, we set the value of $c = {}^1/_2$. For AMSGrad, we compare against the variant employing the Armijo SLS with $c = {}^1/_2$.[3] and use the default (in PyTorch) momentum parameter of $\beta = 0.9$. In Fig. 1, we observe a large variance across step-sizes and poor performance of the default step-size. The best performing variant of AdaGrad/AMSGrad has a step-size of order $10^2$. The line-search variants have good performance across margins, often better than the best-performing constant step-size.

---

[3]This corresponds to the largest allowable step-size in Theorem 4 *without momentum*. Unfortunately, the values of $c$ suggested by the analysis incorporating momentum Theorem 5 are too conservative.

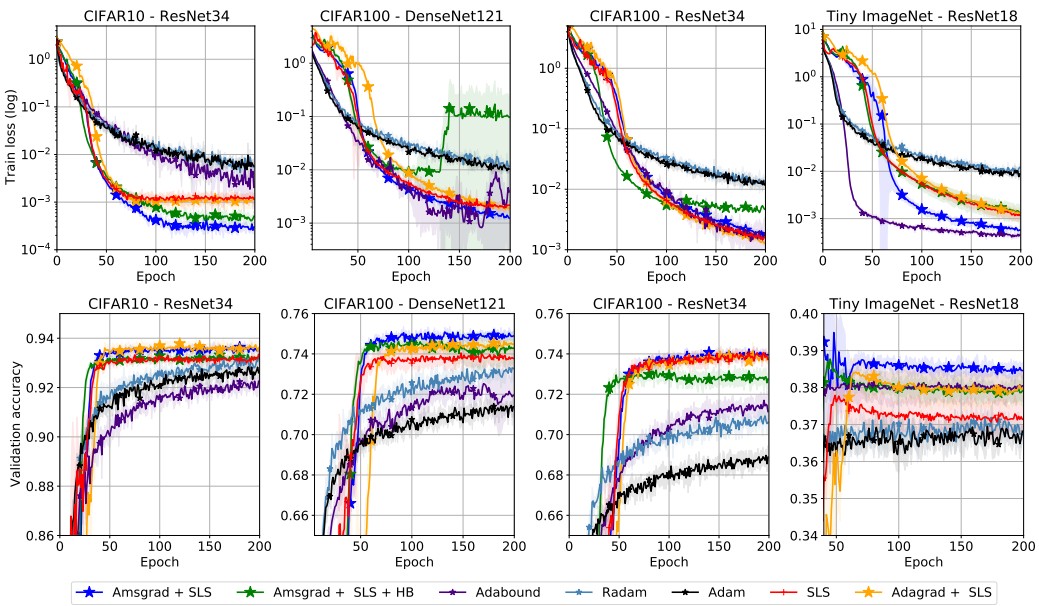

Figure 2: Comparing optimizers for multi-class classification with deep networks. Training loss (top) and validation accuracy (bottom) for CIFAR-10, CIFAR-100 and Tiny ImageNet.

**Real experiments:** Following the protocol in (Luo et al., 2019; Vaswani et al., 2019b; Loizou et al., 2020), we consider training standard neural network architectures for multi-class classification on CIFAR-10, CIFAR-100 and variants of the ImageNet datasets. For each of these experiments, we use a batch-size of 128 and compare against Adam with the best constant step-size found by grid-search. We also include recent improved variants of Adam; RAdam (Liu et al., 2020) and AdaBound (Luo et al., 2019). To see the effect of preconditioning, we compare against SGD with SLS (Vaswani et al., 2019a) and SPS (Loizou et al., 2020). We find that SGD with SLS is more stable and has consistently better test performance than SPS, and hence we only show results for SLS. We also compared against tuned constant step-size SGD and similar to (Vaswani et al., 2019a), we observe that it is consistently outperformed by SGD with SLS.

For the proposed methods, we consider the combinations with theoretical guarantees in the convex setting, specifically AdaGrad and AMSGrad with the Armijo SLS. For AdaGrad, we only show Armijo SLS since it consistently outperforms the Lipschitz line-search. For all variants with Armijo SLS, we use $c = 0.5$ for all convex experiments (suggested by Theorem 4 and Vaswani et al. (2019a)). Since we do not have a theoretical analysis for non-convex problems, we follow the protocol in Vaswani et al. (2019a) and set $c = 0.1$ for all the non-convex experiments. Throughout, we set $\beta = 0.9$ for AMSGrad. We also compare to the AMSGrad variant with heavy-ball (HB) momentum (with $\gamma = 0.25$ found by grid-search). We refer to Appendix F for a detailed discussion about the practical considerations and pseudocodes for the SLS/SPS variants.

We show a subset of results for CIFAR-10, CIFAR-100 and Tiny ImageNet and defer the rest to Appendix G. From Fig. 2 we make the following observations, (i) in terms of generalization, AdaGrad and AMSGrad with Armijo SLS have consistently the best performance, while SGD with SLS is often competitive. (ii) the AdaGrad and AMSGrad variants not only converge faster than Adam and Radam but also with considerably better test performance. AdaBound has comparable convergence in terms of training loss, but does not generalize as well. (iii) AMSGrad momentum is consistently better than the heavy-ball (HB) variant. Moreover, we observed that HB momentum was quite sensitive to the setting of $\gamma$, whereas AMSGrad is robust to $\beta$. In Appendix G, we include ablation results for AMSGrad with Armijo SLS but *without* momentum, and conclude that momentum does indeed improve the performance. In Appendix G, we plot the wall-clock time for the SLS variants and verify that the performance gains justify the increase in wall-clock time per epoch. In the appendix, we show the variation of step-size across epochs, observing a warm-up phase where the step-size increases followed by a constant or decreasing step-size (Goyal et al., 2017).

In Appendix G, we also consider binary classification with RBF kernels for datasets from LIB-SVM (Chang & Lin, 2011) and study the effect of over-parameterization for deep matrix factorization (Rolinek & Martius, 2018; Vaswani et al., 2019b). We show that the same trends hold across different datasets, deep models, deep matrix factorization, and binary classification using kernels.

Our results indicate that simply setting the correct step-size on the fly can lead to substantial empirical gains, often more than those obtained by designing a different preconditioner. Furthermore, we see that with an appropriate step-size adaptation, adaptive gradient methods can generalize better than SGD. By disentangling the effect of the step-size from the preconditioner, our results show that AdaGrad has good empirical performance, contradicting common knowledge. Moreover, our techniques are orthogonal to designing better preconditioners and can be used with other adaptive gradient or even second-order methods.

## 6 DISCUSSION

When training over-parameterized models in the interpolation setting, we showed that for smooth, convex functions, constant step-size variants of both AdaGrad and AMSGrad are guaranteed to converge to the minimizer at $\mathcal{O}(1/T)$ rates. We proposed to use stochastic line-search techniques to help these methods adapt to the function's local smoothness, alleviating the need to tune their step-size and resulting in consistent empirical improvements across tasks. Although adaptive gradient methods outperform SGD in practice, their convergence rates are worse than constant step-size SGD and we hope to address this discrepancy in the future.

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

# Supplementary material

ORGANIZATION OF THE APPENDIX

| Step-size | Rate | Reference |
|---|---|---|
| Constant | $\mathcal{O}(1/T + \sigma/\sqrt{T})$ | Theorem 1 |
| Conservative Lipschitz LS | $\mathcal{O}(1/T + \sigma/\sqrt{T})$ | Theorem 2 |
| Non-conservative LS (with interpolation) | $\mathcal{O}(1/T)$ | Theorem 7 |

| | | |
|---|---|---|
| Constant | $\mathcal{O}(1/T + \sigma^2)$ | Theorem 8 |
| Armijo LS | $\mathcal{O}(1/T + \sigma^2)$ | Theorem 4 |

| | | |
|---|---|---|
| Constant | $\mathcal{O}(1/T + \sigma^2)$ | Theorem 3 |
| Conservative Armijo LS | $\mathcal{O}(1/T + \sigma^2)$ | Theorem 10 |
| Conservative Armijo SPS | $\mathcal{O}(1/T + \sigma^2)$ | Theorem 5 |

Proofs for AMSGrad with heavy ball momentum

| | | |
|---|---|---|
| Constant | $\mathcal{O}(1/T + \sigma^2)$ | Theorem 11 |
| Conservative Armijo LS | $\mathcal{O}(1/T + \sigma^2)$ | Theorem 13 |
| Conservative Armijo SPS | $\mathcal{O}(1/T + \sigma^2)$ | Theorem 12 |

Table 3: Summary of notation

| Concept | Symbol | Concept | Symbol |
|---|---|---|---|
| Iteration counter, maximum | $k, T$ | General preconditioner | $A_k$ |
| Iterates, minimum | $w_k, w^*$ | Preconditioner bounds | $[a_{\min}, a_{\max}]$ |
| Step-size | $\eta_k$ | Maximum smoothness | $L_{\max}$ |
| Function value, minimum | $f(w), f^*$ | Dimensionality | $d$ |
| Stoch. function value, minimum | $f_i(w), f_i^*$ | Diameter bound | $D$ |
| | | Variance | $\sigma^2 = \mathbb{E}_i[f_i(w^*) - f_i^*]$ |

## A  SETUP AND ASSUMPTIONS

We restate the main notation in Table 3. We now restate the main assumptions required for our theoretical results

We assume our objective $f : \mathbb{R}^d \to \mathbb{R}$ has a finite-sum structure,

$$f(w) = \frac{1}{n} \sum_{i=1}^{n} f_i(w), \qquad (4)$$

and analyze the following update, with $i_k$ selected uniformly at random,

$$w_{k+1} = w_k - \eta_k A_k^{-1} m_k \quad ; \quad m_k = \beta m_{k-1} + (1 - \beta) \nabla f_{i_k}(w_k) \qquad \text{(Update rule)}$$

where $\eta_k$ is either a pre-specified constant or selected on the fly. We consider AdaGrad and AMSGrad and use the fact that the preconditioners are non-decreasing i.e. $A_k \succeq A_{k-1}$. For AdaGrad, $\beta = 0$. For AMSGrad, we further assume that the preconditioners remain bounded with eigenvalues in the range $[a_{\min}, a_{\max}]$,

$$a_{\min} I \preceq A_k \preceq a_{\max} I. \qquad \text{(Bounded preconditioner)}$$

For all algorithms, we assume that the iterates do not diverge and remain in a ball of radius $D$, as is standard in the literature on online learning (Duchi et al., 2011; Levy et al., 2018) and adaptive gradient methods (Reddi et al., 2018),

$$\|w_k - w^*\| \leq D. \qquad \text{(Bounded iterates)}$$

Our main assumptions are that each individual function $f_i$ is convex, differentiable, has a finite minimum $f_i^*$, and is $L_i$-smooth, meaning that for all $v$ and $w$,

$$f_i(v) \geq f_i(w) - \langle \nabla f_i(w), w - v \rangle, \qquad \text{(Individual Convexity)}$$

$$f_i(v) \leq f_i(w) + \langle \nabla f_i(w), v - w \rangle + \frac{L_i}{2} \|v - w\|^2, \qquad \text{(Individual Smoothness)}$$

which also implies that $f$ is convex and $L_{\max}$-smooth, where $L_{\max}$ is the maximum smoothness constant of the individual functions. A consequence of smoothness is the following bound on the norm of the gradient stochastic gradients,

$$\|\nabla f_i(w)\|^2 \leq 2 L_{\max}(f_i(w) - f_i^*).$$

To characterize interpolation, we define the expected difference between the minimum of $f$, $f(w^*)$, and the minimum of the individual functions $f_i^*$,

$$\sigma^2 = \mathbb{E}_i[f_i(w^*) - f_i^*] < \infty. \qquad \text{(Noise)}$$

When interpolation is exactly satisfied, every data point can be fit exactly, such that $f_i^* = 0$ and $f(w^*) = 0$, we have $\sigma^2 = 0$.

# B   LINE-SEARCH AND POLYAK STEP-SIZES

We now give the main guarantees on the step-sizes returned by the line-search. In practice, we use a backtracking line-search to find a step-size that satisfies the constraints, described in Algorithm 1 (Appendix F). For simplicity of presentation, here we assume the line-search returns the largest step-size that satisfies the constraints.

When interpolation is not exactly satisfied, the procedures need to be equipped with an additional safety mechanism; either by capping the maximum step-size by some $\eta_{\max}$ or by ensuring non-increasing step-sizes, $\eta_k \le \eta_{k-1}$. In this case, $\eta_{\max}$ ensures that a bad iteration of the line-search procedure does not result in divergence. When interpolation is satisfied, those conditions can be dropped (e.g., setting $\eta_{\max} \to \infty$) and the rate does not depend on $\eta_{\max}$. The line-searches depend on a parameter $c \in (0,1)$ that controls how much decrease is necessary to accept a step (larger $c$ means more decrease is demanded).

Assuming the Lipschitz and Armijo line-searches select the largest $\eta$ such that

$$f_i(w - \eta \nabla f_i(w)) \le f_i(w) - c\eta \left\| \nabla f_i(w) \right\|^2, \qquad \eta \le \eta_{\max}, \qquad \text{(Lipschitz line-search)}$$

$$f_i(w - \eta A^{-1} \nabla f_i(w)) \le f_i(w) - c\eta \left\| \nabla f_i(w) \right\|_{A^{-1}}^2, \qquad \eta \le \eta_{\max}, \qquad \text{(Armijo line-search)}$$

the following lemma holds.

> **Lemma 1** (Line-search). *If $f_i$ is $L_i$-smooth, the Lipschitz and Armijo lines-searches ensure*
>
> $$\eta \left\| \nabla f_i(w) \right\|^2 \le \frac{1}{c}(f_i(w) - f_i^*), \quad and \qquad \min \left\{ \eta_{\max}, \frac{2\,(1-c)}{L_i} \right\} \le \eta \le \eta_{\max},$$
>
> $$\eta \left\| \nabla f_i(w) \right\|_{A^{-1}}^2 \le \frac{1}{c}(f_i(w) - f_i^*), \quad and \quad \min \left\{ \eta_{\max}, \frac{2\,\lambda_{\min}(A)\,(1-c)}{L_i} \right\} \le \eta \le \eta_{\max}.$$

We do not include the backtracking line-search parameters in the analysis for simplicity, as the same bounds hold, up to some constant. With a backtracking line-search, we start with a large enough candidate step-size and multiply it by some constant $\gamma < 1$ until the Lipschitz or Armijo line-search condition is satisfied. If $\eta'$ was a proposal step-size that did not satisfy the constraint, but $\gamma \eta'$ does, the maximum step-size $\eta$ that satisfies the constraint must be in the range $\gamma \eta' \le \eta < \eta'$.

*Proof of Lemma 1.* Recall that if $f_i$ is $L_i$-smooth, then for an arbitrary direction $d$,

$$f_i(w - d) \le f_i(w) - \langle \nabla f_i(w), d \rangle + \frac{L_i}{2} \left\| d \right\|^2.$$

For the Lipschitz line-search, $d = \eta \nabla f_i(w)$. The smoothness and the line-search condition are then

Smoothness: $\qquad f_i(w - \eta \nabla f_i(w)) - f_i(w) \le \left( \frac{L_i}{2}\eta^2 - \eta \right) \left\| \nabla f_i(w) \right\|^2,$

Line-search: $\qquad f_i(w - \eta \nabla f_i(w)) - f_i(w) \le -c\eta \left\| \nabla f_i(w) \right\|^2.$

As illustrated in Fig. 3, the line-search condition is looser than smoothness if

$$\left( \tfrac{L_i}{2}\eta^2 - \eta \right) \left\| \nabla f_i(w) \right\|^2 \le -c\eta \left\| \nabla f_i(w) \right\|^2.$$

The inequality is satisfied for any $\eta \in [a,b]$, where $a, b$ are values of $\eta$ that satisfy the equation with equality, $a = 0, b = 2(1-c)/L_i$, and the line-search condition holds for $\eta \le 2(1-c)/L_i$.

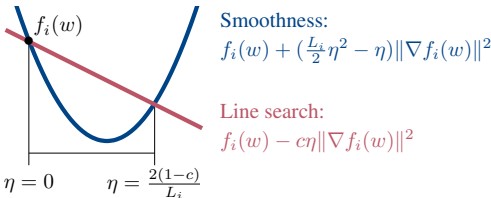

Figure 3: Sketch of the line-search inequalities.

As the line-search selects the largest feasible step-size, $\eta \ge 2(1-c)/L_i$. If the step-size is capped at $\eta_{\max}$, we have $\eta \ge \min\{\eta_{\max}, 2(1-c)/L_i\}$, and the proof for the Lipschitz line-search is complete. The proof for the Armijo line-search is identical except for the smoothness property, which is modified

to use the $\|\cdot\|_A$-norm for the direction $d = \eta A^{-1} \nabla f_i(w)$;

$$f_i(w - \eta A^{-1} \nabla f_i(w)) \leq f_i(w) - \eta \langle \nabla f_i(w), A^{-1} \nabla f_i(w) \rangle + \frac{L_i}{2} \eta^2 \left\| A^{-1} \nabla f_i(w) \right\|^2,$$

$$\leq f_i(w) - \eta \left\| \nabla f_i(w) \right\|_{A^{-1}}^2 + \frac{L_i}{2\lambda_{\min}(A)} \eta^2 \left\| \nabla f_i(w) \right\|_{A^{-1}}^2,$$

$$= f_i(w) + \left( \frac{L_i}{2\lambda_{\min}(A)} \eta^2 - \eta \right) \left\| \nabla f_i(w) \right\|_{A^{-1}}^2,$$

where the second inequality comes from $\|A^{-1} \nabla f_i(w)\|^2 \leq \frac{1}{\lambda_{\min}(A)} \|\nabla f_i(w)\|_{A^{-1}}^2$. $\qquad\square$

Similarly, the stochastic Polyak step-sizes (SPS) for $f_i$ at $w$ are defined as

$$\text{SPS:} \quad \eta = \min \left\{ \frac{f_i(w) - f_i^*}{c \left\| \nabla f_i(w) \right\|^2}, \eta_{\max} \right\}, \quad \text{Armijo SPS:} \quad \eta = \min \left\{ \frac{f_i(w) - f_i^*}{c \left\| \nabla f_i(w) \right\|_{A^{-1}}^2}, \eta_{\max} \right\},$$

where the parameter $c > 0$ controls the scaling of the step (larger $c$ means smaller steps).

---

**Lemma 2** (SPS guarantees). *If $f_i$ is $L_i$-smooth, SPS and Armijo SPS ensure that*

$$\text{SPS:} \qquad \eta \left\| \nabla f_i(w) \right\|^2 \leq \tfrac{1}{c}(f_i(w) - f_i^*), \qquad \min \left\{ \eta_{\max}, \tfrac{1}{2cL_i} \right\} \leq \eta \leq \eta_{\max},$$

$$\text{Armijo SPS:} \quad \eta \left\| \nabla f_i(w) \right\|_{A^{-1}}^2 \leq \tfrac{1}{c}(f_i(w) - f_i^*), \quad \min \left\{ \eta_{\max}, \tfrac{\lambda_{\min}(A)}{2cL_i} \right\} \leq \eta \leq \eta_{\max}$$

---

*Proof of Lemma 2.* The first guarantee follows directly from the definition of the step-size. For SPS,

$$\eta \left\| \nabla f_i(w) \right\|^2 = \min \left\{ \frac{f_i(w) - f_i^*}{c \left\| \nabla f_i(w) \right\|^2}, \eta_{\max} \right\} \left\| \nabla f_i(w) \right\|^2,$$

$$= \min \left\{ \frac{f_i(w) - f_i^*}{c}, \eta_{\max} \left\| \nabla f_i(w) \right\|^2 \right\} \leq \frac{1}{c}(f_i(w) - f_i^\star).$$

The same inequalities hold for Armijo SPS with $\|\nabla f_i(w)\|_{A^{-1}}^2$. To lower-bound the step-size, we use the $L_i$-smoothness of $f_i$, which implies $f_i(w) - f_i^* \geq \frac{1}{2L_i} \|\nabla f_i(w)\|^2$. For SPS,

$$\frac{f_i(w) - f_i^*}{c \left\| \nabla f_i(w) \right\|^2} \geq \frac{\frac{1}{2L_i} \left\| \nabla f_i(w) \right\|^2}{c \left\| \nabla f_i(w) \right\|^2} = \frac{1}{2cL_i}.$$

For Armijo SPS, we additionally use $\|\nabla f_i(w)\|_{A^{-1}}^2 \leq \frac{1}{\lambda_{\min}(A)} \|\nabla f_i(w)\|^2$,

$$\frac{f_i(w) - f_i^*}{c \left\| \nabla f_i(w) \right\|_{A^{-1}}^2} \geq \frac{\frac{1}{2L_i} \left\| \nabla f_i(w) \right\|^2}{c \frac{1}{\lambda_{\min}(A)} \left\| \nabla f_i(w) \right\|^2} = \frac{\lambda_{\min}(A)}{2cL_i}. \qquad\square$$

## C    PROOFS FOR ADAGRAD

We now move to the proof of the convergence of AdaGrad in the smooth setting with a constant step-size (Theorem 1) and the conservative Lipschitz line-search (Theorem 2). We first give a rate for an arbitrary step-size $\eta_k$ in the range $[\eta_{\min}, \eta_{\max}]$, and derive the rates of Theorems 1 and 2 by specializing the range to a constant step-size or line-search.

---

**Proposition 1** (AdaGrad with non-increasing step-sizes). *Assuming (i) convexity and (ii) $L_{\max}$-smoothness of each $f_i$, and (iii) bounded iterates, AdaGrad with non-increasing ($\eta_k \leq \eta_{k-1}$), bounded step-sizes ($\eta_k \in [\eta_{\min}, \eta_{\max}]$), and uniform averaging $\bar{w}_T = \frac{1}{T} \sum_{k=1}^{T} w_k$, converges at a rate*

$$\mathbb{E}[f(\bar{w}_T) - f^*] \leq \frac{\alpha}{T} + \frac{\sqrt{\alpha}\sigma}{\sqrt{T}}, \qquad \text{where } \alpha = \frac{1}{2}\left(\frac{D^2}{\eta_{\min}} + 2\eta_{\max}\right)^2 dL_{\max}.$$

---

We first use the above result to prove Theorems 1 and 2. The proof of Theorem 1 is immediate by plugging $\eta = \eta_{\min} = \eta_{\max}$ in Proposition 1. We recall its statement;

---

**Theorem 1** (Constant step-size AdaGrad). *Assuming (i) convexity and (ii) $L_{\max}$-smoothness of each $f_i$, and (iii) bounded iterates, AdaGrad with a constant step-size $\eta$ and uniform averaging such that $\bar{w}_T = \frac{1}{T}\sum_{k=1}^{T} w_k$, converges at a rate*

$$\mathbb{E}[f(\bar{w}_T) - f^*] \leq \frac{\alpha}{T} + \frac{\sqrt{\alpha}\sigma}{\sqrt{T}}, \quad \text{where } \alpha = \frac{1}{2}\left(\frac{D^2}{\eta} + 2\eta\right)^2 dL_{\max}.$$

---

For Theorem 2, we use the properties of the conservative Lipschitz line-search. We recall its statement;

---

**Theorem 2.** *Under the same assumptions as Theorem 1, AdaGrad with a conservative Lipschitz line-search with $c = 1/2$, a step-size upper bound $\eta_{\max}$ and uniform averaging converges at a rate*

$$\mathbb{E}[f(\bar{w}_T) - f^*] \leq \frac{\alpha}{T} + \frac{\sqrt{\alpha}\sigma}{\sqrt{T}}, \quad \text{where } \alpha = \frac{1}{2}\left(D^2 \max\left\{\frac{1}{\eta_{\max}}, L_{\max}\right\} + 2\,\eta_{\max}\right)^2 dL_{\max}.$$

---

*Proof of Theorem 2.* Using Lemma 1, there is a step-size $\eta_k$ that satisfies the Lipschitz line-search with $\eta_k \geq 2(1-c)/L_{\max}$. Setting $c = 1/2$ and using a maximum step-size $\eta_{\max}$, we have

$$\min\left\{\eta_{\max}, \frac{1}{L_{\max}}\right\} \leq \eta_k \leq \eta_{\max}, \qquad \Longrightarrow \qquad \frac{1}{\eta_{\min}} = \max\left\{\frac{1}{\eta_{\max}}, L_{\max}\right\}. \qquad \square$$

Before going into the proof of Proposition 1, we recall some standard lemmas from the adaptive gradient literature (Theorem 7 & Lemma 10 in (Duchi et al., 2011), Lemma 5.15 & 5.16 in (Hazan, 2016)), and a useful quadratic inequality (Levy et al., 2018, Part of Theorem 4.2)). We include proofs in Appendix C.1 for completeness.

---

**Lemma 3.** *If the preconditioners are non-decreasing ($A_k \succeq A_{k-1}$), the step-sizes are non-increasing ($\eta_k \leq \eta_{k-1}$), and the iterates stay within a ball of radius $D$ of the minima,*

$$\sum_{k=1}^{T} \|w_k - w^*\|^2_{\frac{1}{\eta_k} A_k - \frac{1}{\eta_{k-1}} A_{k-1}} \leq \frac{D^2}{\eta_T}\mathsf{Tr}(A_T).$$

---

**Lemma 4.** *For AdaGrad, $A_k = \left[\sum_{i=1}^{k} \nabla f_{i_k}(w_k)\nabla f_{i_k}(w_k)^\top\right]^{1/2}$ and satisfies,*

$$\sum_{k=1}^{T} \|\nabla f_{i_k}(w_k)\|^2_{A_k^{-1}} \leq 2\mathsf{Tr}(A_T), \qquad \mathsf{Tr}(A_T) \leq \sqrt{d\sum_{k=1}^{T} \|\nabla f_{i_k}(w_k)\|^2}.$$

---

**Lemma 5.** *If $x^2 \leq a(x + b)$ for $a \geq 0$ and $b \geq 0$,*

$$x \leq \frac{1}{2}\left(\sqrt{a^2 + 4ab} + a\right) \leq a + \sqrt{ab}.$$

We now prove Proposition 1.

*Proof of Proposition 1.* We first give an overview of the main steps. Using the definition of the update rule, along with Lemmas 3 and 4, we will show that

$$2\sum_{k=1}^{T}\langle\nabla f_{i_k}(w_k), w_k - w^*\rangle \leq \left(\frac{D^2}{\eta_{\min}} + 2\eta_{\max}\right)\mathsf{Tr}(A_T). \tag{5}$$

Using the definition of $A_T$, individual smoothness and convexity, we then show that for a constant $a$,

$$\sum_{k=1}^{T}\mathbb{E}[f(w_k) - f^*] \leq a\left(\mathbb{E}\left[\sqrt{\sum_{k=1}^{T}f_{i_k}(w_k) - f_{i_k}(w^*)}\right] + T\sigma^2\right), \tag{6}$$

Using the quadratic inequality (Lemma 5), averaging and using Jensen's inequality finishes the proof.

To derive Eq. (5), we start with the Update rule, measuring distances to $w^*$ in the $\|\cdot\|_{A_k}$ norm,

$$\|w_{k+1} - w^*\|_{A_k}^2 = \|w_k - w^*\|_{A_k}^2 - 2\eta_k\langle\nabla f_{i_k}(w_k), w_k - w^*\rangle + \eta_k^2\|\nabla f_{i_k}(w_k)\|_{A_k^{-1}}^2.$$

Dividing by $\eta_k$, reorganizing the equation and summing across iterations yields

$$2\sum_{k=1}^{T}\langle\nabla f_{i_k}(w_k), w_k - w^*\rangle \leq \sum_{k=1}^{T}\|w_k - w^*\|_{\left(\frac{A_k}{\eta_k} - \frac{A_{k-1}}{\eta_{k-1}}\right)}^2 + \sum_{k=1}^{T}\eta_k\|\nabla f_{i_k}(w_k)\|_{A_k^{-1}}^2,$$

$$\leq \sum_{k=1}^{T}\|w_k - w^*\|_{\left(\frac{A_k}{\eta_k} - \frac{A_{k-1}}{\eta_{k-1}}\right)}^2 + \eta_{\max}\sum_{k=1}^{T}\|\nabla f_{i_k}(w_k)\|_{A_k^{-1}}^2.$$

We use the Lemmas 3, 4 to bound the RHS by the trace of the last preconditioner,

$$\leq \frac{D^2}{\eta_T}\mathsf{Tr}(A_T) + 2\eta_{\max}\mathsf{Tr}(A_T), \qquad\qquad \text{(Lemmas 3 and 4)}$$

$$\leq \left(\frac{D^2}{\eta_{\min}} + 2\eta_{\max}\right)\mathsf{Tr}(A_T). \qquad\qquad (\eta_k \geq \eta_{\min})$$

To derive Eq. (6), we bound the trace of $A_T$ using Lemma 4 and Individual Smoothness,

$$\mathsf{Tr}(A_T) \leq \sqrt{d}\sqrt{\sum_{k=1}^{T}\|\nabla f_{i_k}(w_k)\|^2}, \qquad\qquad \text{(Lemma 4, Trace bound)}$$

$$\leq \sqrt{2dL_{\max}}\sqrt{\sum_{k=1}^{T}f_{i_k}(w_k) - f_{i_k}^*}. \qquad\qquad \text{(Individual Smoothness)}$$

$$\leq \sqrt{2dL_{\max}}\sqrt{\sum_{k=1}^{T}f_{i_k}(w_k) - f_{i_k}(w^*) + f_{i_k}(w^*) - f_{i_k}^*} \qquad (\pm f_{i_k}(w^*))$$

Combining the above inequalities with $\delta_{i_k} = f_{i_k}(w^*) - f_{i_k}^*$ and $a = \frac{1}{2}(\frac{D^2}{\eta_{\min}} + 2\eta_{\max})\sqrt{2dL_{\max}}$,

$$\sum_{k=1}^{T}\langle\nabla f_{i_k}(w_k), w_k - w^*\rangle \leq a\sqrt{\sum_{k=1}^{T}f_{i_k}(w_k) - f_{i_k}(w^*) + \delta_{i_k}}.$$

Using Individual Convexity and taking expectations,

$$\sum_{k=1}^{T}\mathbb{E}[f(w_k) - f^*] \leq a\,\mathbb{E}\left[\sqrt{\sum_{k=1}^{T}f_{i_k}(w_k) - f_{i_k}(w^*) + \delta_{i_k}}\right],$$

$$\leq a\sqrt{\mathbb{E}\left[\sum_{k=1}^{T}f_{i_k}(w_k) - f_{i_k}(w^*) + \delta_{i_k}\right]}. \qquad \text{(Jensen's inequality)}$$

Letting $\sigma^2 := \mathbb{E}_i[\delta_i] = \mathbb{E}_i[f_i(w^*) - f_i^*]$ and taking the square on both sides yields

$$\left(\sum_{k=1}^{T}\mathbb{E}[f(w_k) - f^*]\right)^2 \leq a^2\left(\mathbb{E}\left[\sum_{k=1}^{T}f_{i_k}(w_k) - f_{i_k}(w^*)\right] + T\sigma^2\right).$$

The quadratic bound (Lemma 5) $x^2 \leq \alpha(x + \beta)$ implies $x \leq \alpha + \sqrt{\alpha\beta}$, with

$$x = \sum_{k=1}^{T} \mathbb{E}[f(w_k) - f^*], \qquad \alpha = \frac{1}{2}\left(D^2 \frac{1}{\eta_{\min}} + 2\eta_{\max}\right)^2 dL_{\max}, \qquad \beta = T\sigma^2,$$

gives the first bound below. Averaging $\bar{w}_T = \frac{1}{T}\sum_{k=1}^{T} w_k$ and using Jensen's inequality give the result;

$$\sum_{k=1}^{T} \mathbb{E}[f(w_k) - f^*] \leq \alpha + \sqrt{\alpha\beta} \qquad \implies \qquad \mathbb{E}[f(\bar{w}_T) - f^*] \leq \frac{\alpha}{T} + \frac{\sqrt{\alpha}\sigma}{\sqrt{T}}. \qquad \square$$

### C.1 PROOFS OF ADAPTIVE GRADIENT LEMMAS

For completeness, we give proofs for the lemmas used in the previous section. We restate them here;

---

**Lemma 3.** *If the preconditioners are non-decreasing ($A_k \succeq A_{k-1}$), the step-sizes are non-increasing ($\eta_k \leq \eta_{k-1}$), and the iterates stay within a ball of radius $D$ of the minima,*

$$\sum_{k=1}^{T} \|w_k - w^*\|^2_{\frac{1}{\eta_k} A_k - \frac{1}{\eta_{k-1}} A_{k-1}} \leq \frac{D^2}{\eta_T} \mathsf{Tr}(A_T).$$

---

*Proof of Lemma 3.* Under the assumptions that $A_k$ is non-decreasing and $\eta_k$ is non-increasing, $\frac{1}{\eta_k} A_k - \frac{1}{\eta_{k-1}} A_{k-1} \succeq 0$, so we can use the Bounded iterates assumption to bound

$$\sum_{k=1}^{T} \|w_k - w^*\|^2_{\frac{A_k}{\eta_k} - \frac{A_{k-1}}{\eta_{k-1}}} \leq \sum_{k=1}^{T} \lambda_{\max}\left(\frac{A_k}{\eta_k} - \frac{A_{k-1}}{\eta_{k-1}}\right) \|w_k - w^*\|^2$$

$$\leq D^2 \sum_{k=1}^{T} \lambda_{\max}\left(\frac{A_k}{\eta_k} - \frac{A_{k-1}}{\eta_{k-1}}\right).$$

We then upper-bound $\lambda_{\max}$ by the trace and use the linearity of the trace to telescope the sum,

$$\leq D^2 \sum_{k=1}^{T} \mathsf{Tr}\left(\frac{A_k}{\eta_k} - \frac{A_{k-1}}{\eta_{k-1}}\right) = D^2 \sum_{k=1}^{T} \mathsf{Tr}\left(\frac{A_k}{\eta_k}\right) - \mathsf{Tr}\left(\frac{A_{k-1}}{\eta_{k-1}}\right),$$

$$= D^2 \left(\mathsf{Tr}\left(\frac{A_T}{\eta_T}\right) - \mathsf{Tr}\left(\frac{A_0}{\eta_0}\right)\right) \leq D^2 \frac{1}{\eta_T} \mathsf{Tr}(A_T). \qquad \square$$

---

**Lemma 4.** *For AdaGrad, $A_k = \left[\sum_{i=1}^{k} \nabla f_{i_k}(w_k) \nabla f_{i_k}(w_k)^\top\right]^{1/2}$ and satisfies,*

$$\sum_{k=1}^{T} \|\nabla f_{i_k}(w_k)\|^2_{A_k^{-1}} \leq 2\mathsf{Tr}(A_T), \qquad \mathsf{Tr}(A_T) \leq \sqrt{d \sum_{k=1}^{T} \|\nabla f_{i_k}(w_k)\|^2}.$$

---

*Proof of Lemma 4.* For ease of notation, let $\nabla_k := \nabla f_{i_k}(w_k)$. By induction, starting with $T = 1$,

$$\|\nabla f_{i_1}(w_1)\|^2_{A_1^{-1}} = \nabla_1^\top A_1^{-1} \nabla_1 = \mathsf{Tr}\left(\nabla_1^\top A_1^{-1} \nabla_1\right) = \mathsf{Tr}\left(A_1^{-1} \nabla_1 \nabla_1^\top\right), \quad \text{(Cyclic property of trace)}$$

$$= \mathsf{Tr}\left(A_1^{-1} A_1^2\right) = \mathsf{Tr}(A_1). \qquad (A_1 = (\nabla_1 \nabla_1^\top)^{1/2})$$

Suppose that it holds for $T - 1$, $\sum_{k=1}^{T-1} \|\nabla_k\|^2_{A_k^{-1}} \leq 2\mathsf{Tr}(A_{T-1})$. We will show that it also holds for $T$. Using the definition of the preconditioner and the cyclic property of the trace,

$$\sum_{k=1}^{T} \|\nabla f_{i_k}(w_k)\|^2_{A_k^{-1}} \leq 2\mathsf{Tr}(A_{T-1}) + \|\nabla_T\|^2_{A_T^{-1}} \qquad \text{(Induction hypothesis)}$$

$$= 2\mathsf{Tr}\left((A_T^2 - \nabla_T \nabla_T^\top)^{1/2}\right) + \mathsf{Tr}\left(A_T^{-1} \nabla_T \nabla_T^\top\right) \qquad \text{(AdaGrad update)}$$

We then use the fact that for any $X \succeq Y \succeq 0$, we have (Duchi et al., 2011, Lemma 8)

$$2\mathsf{Tr}\left((X - Y)^{1/2}\right) + \mathsf{Tr}\left(X^{-1/2} Y\right) \leq 2\mathsf{Tr}\left(X^{1/2}\right).$$

As $X = A_T^2 \succeq Y = \nabla_T \nabla_T^\top \succeq 0$, we can use the above inequality and the induction holds for $T$.

For the trace bound, recall that $A_T = G_T^{1/2}$ where $G_T = \sum_{i=1}^{T} \nabla f_{i_k}(w_k) \nabla f_{i_k}(w_k)^\top$. We use Jensen's inequality,

$$\mathsf{Tr}(A_T) = \mathsf{Tr}\left(G_T^{1/2}\right) = \sum_{j=1}^{d} \sqrt{\lambda_j(G_T)} = d\left(\frac{1}{d} \sum_{j=1}^{d} \sqrt{\lambda_j(G_T)}\right),$$

$$\leq d\sqrt{\frac{1}{d} \sum_{j=1}^{d} \lambda_j(G_T)} = \sqrt{d}\sqrt{\mathsf{Tr}(G_T)}.$$

To finish the proof, we use the definition of $G_T$ and the linearity of the trace to get

$$\sqrt{\mathsf{Tr}(G_T)} = \sqrt{\mathsf{Tr}\left(\sum_{k=1}^{T} \nabla_k \nabla_k^\top\right)} = \sqrt{\sum_{k=1}^{T} \mathsf{Tr}(\nabla_k \nabla_k^\top)} = \sqrt{\sum_{k=1}^{T} \|\nabla_k\|^2}. \qquad \square$$

**Lemma 5.** *If $x^2 \leq a(x + b)$ for $a \geq 0$ and $b \geq 0$,*

$$x \leq \frac{1}{2}\left(\sqrt{a^2 + 4ab} + a\right) \leq a + \sqrt{ab}.$$

*Proof of Lemma 5.* The starting point is the quadratic inequality $x^2 - ax - ab \leq 0$. Letting $r_1 \leq r_2$ be the roots of the quadratic, the inequality holds if $x \in [r_1, r_2]$. The upper bound is then given by using $\sqrt{a + b} \leq \sqrt{a} + \sqrt{b}$

$$r_2 = \frac{a + \sqrt{a^2 + 4ab}}{2} \leq \frac{a + \sqrt{a^2} + \sqrt{4ab}}{2} = a + \sqrt{ab}. \qquad \square$$

## C.2 REGRET BOUND FOR ADAGRAD UNDER INTERPOLATION

In the online convex optimization framework, we consider a sequence of functions $f_k|_{k=1}^T$, chosen potentially adversarially by the environment. The aim of the learner is to output a series of strategies $w_k|_{k=1}^T$ *before* seeing the function $f_k$. After choosing $w_k$, the learner suffers the loss $f_k(w_k)$ and observes the corresponding gradient vector $\nabla f_k(w_k)$. They suffer an instantaneous regret $r_k = f_k(w_k) - f_k(w)$ compared to a fixed strategy $w$. The aim is to bound the cumulative regret,

$$R(T) = \sum_{k=1}^T [f_k(w_k) - f_k(w^*)]$$

where $w^* = \arg\min \sum_{k=1}^T f_k(w)$ is the best strategy if we had access to the entire sequence of functions in hindsight. Assuming the functions are convex but non-smooth, AdaGrad obtains an $\mathcal{O}(1/\sqrt{T})$ regret bound (Duchi et al., 2011). For online convex optimization, the interpolation assumption implies that the learner model is powerful enough to fit the entire sequence of functions. For large over-parameterized models like neural networks, where the number of parameters is of the order of millions, this is a reasonable assumption for large $T$.

We first recall the update of AdaGrad, at iteration $k$, the learner decides to play the strategy $w_k$, suffers loss $f_k(w_k)$ and uses the gradient feedback $\nabla f_k(w_k)$ to update their strategy as

$$w_{k+1} = w_k - \eta A_k^{-1} \nabla f_k(w_k), \quad \text{where } A_k = \left[\sum_{i=1}^k \nabla f_k(w_k)\nabla f_k(w_k)^\top\right]^{1/2}.$$

Now we show that for smooth, convex functions under the interpolation assumption, AdaGrad with a constant step-size can result in *constant* regret.

**Theorem 6.** *For a sequence of $L_{\max}$-smooth, convex functions $f_k$, assuming the iterates remain bounded s.t. for all $k$, $\|w_k - w^*\| \leq D$, AdaGrad with a constant step-size $\eta$ achieves the following regret bound,*

$$R(T) \leq \frac{1}{2}\left(D^2 \frac{1}{\eta} + 2\eta\right)^2 dL_{\max} + \sqrt{\frac{1}{2}\left(D^2 \frac{1}{\eta} + 2\eta\right)^2 dL_{\max}\sigma^2} \sqrt{T}$$

*where $\sigma^2$ is an upper-bound on $f_k(w^*) - f_k^*$.*

Observe that $\sigma^2$ is the degree to which interpolation is violated, and if $\sigma^2 \neq 0$, $R(T) = \mathcal{O}(\sqrt{T})$ matching the regret of (Duchi et al., 2011). However, when interpolation is exactly satisfied, $\sigma^2 = 0$, and $R(T) = \mathcal{O}(1)$.

*Proof of Theorem 6.* The proof follows that of Proposition 1 which is inspired from (Levy et al., 2018). For convenience, we repeat the basic steps. Measuring distances to $w^*$ in the $\|\cdot\|_{A_k}$ norm,

$$\|w_{k+1} - w^*\|_{A_k}^2 = \|w_k - w^*\|_{A_k}^2 - 2\eta\langle\nabla f_k(w_k), w_k - w^*\rangle + \eta^2 \|\nabla f_k(w_k)\|_{A_k^{-1}}^2.$$

Dividing by $2\eta$, reorganizing the equation and summing across iterations yields

$$\sum_{k=1}^{T}\langle\nabla f_k(w_k), w_k - w^*\rangle \leq \sum_{k=1}^{T}\|w_k - w^*\|^2_{\left(\frac{A_k}{2\eta} - \frac{A_{k-1}}{2\eta}\right)} + \frac{\eta}{2}\sum_{k=1}^{T}\|\nabla f_k(w_k)\|^2_{A_k^{-1}}.$$

By convexity of $f_k$, $\langle\nabla f_k(w_k), w_k - w^*\rangle \geq f_k(w_k) - f_k(w^*)$. Using the definition of regret,

$$R(T) \leq \sum_{k=1}^{T}\|w_k - w^*\|^2_{\left(\frac{A_k}{2\eta} - \frac{A_{k-1}}{2\eta}\right)} + \frac{\eta}{2}\sum_{k=1}^{T}\|\nabla f_k(w_k)\|^2_{A_k^{-1}}.$$

We use the Lemmas 3, 4 to bound the RHS by the trace of the last preconditioner,

$$R(T) \leq \left(\frac{D^2}{2\eta} + \eta\right)\mathsf{Tr}(A_T).$$

We now bound the trace of $A_T$ using Lemma 4 and Individual Smoothness,

$$\begin{aligned}
\mathsf{Tr}(A_T) &\leq \sqrt{d}\sqrt{\sum_{k=1}^{T}\|\nabla f_k(w_k)\|^2}, &\text{(Lemma 4, Trace bound)}\\
&\leq \sqrt{2dL_{\max}}\sqrt{\sum_{k=1}^{T}f_k(w_k) - f_k^*}, &\text{(Individual Smoothness)}\\
&\leq \sqrt{2dL_{\max}}\sqrt{\sum_{k=1}^{T}f_k(w_k) - f_k(w^*) + f_k(w^*) - f_k^*}, &(\pm f_k(w^*))\\
&\leq \sqrt{2dL_{\max}}\sqrt{R(T) + \sigma^2 T}. &\text{(Since } f_k(w^*) - f_k^* \leq \sigma^2)
\end{aligned}$$

Plugging this back into the regret bound,

$$R(T) \leq \left(\frac{D^2}{2\eta} + \eta\right)\sqrt{2dL_{\max}}[\sqrt{R(T) + \sigma^2 T}].$$

Squaring both sides and denoting $a = \left(\frac{D^2}{2\eta} + \eta\right)\sqrt{2dL_{\max}}$,

$$[R(T)]^2 \leq a^2[R(T) + \sigma^2 T].$$

Using the quadratic bound (Lemma 5) $x^2 \leq \alpha(x + \beta)$ implies $x \leq \alpha + \sqrt{\alpha\beta}$, with

$$x = R(T), \qquad \alpha = \frac{1}{2}\left(D^2\frac{1}{\eta} + 2\eta\right)^2 dL_{\max}, \qquad \beta = \sigma^2 T,$$

yields the bound,

$$R(T) \leq \alpha + \sqrt{\alpha\beta} = \frac{1}{2}\left(D^2\frac{1}{\eta} + 2\eta\right)^2 dL_{\max} + \sqrt{\frac{1}{2}\left(D^2\frac{1}{\eta} + 2\eta\right)^2 dL_{\max}\sigma^2 T}. \qquad \square$$

## C.3 With interpolation, without conservative line-searches

In this section, we show that the conservative constraint $\eta_{k+1} \leq \eta_k$ is not necessary if interpolation is satisfied. We give the proof for the Armijo line-search, that has better empirical performance, but a worse theoretical dependence on the problem's constants. For the theorem below, $a_{\min}$ is lower-bounded by $\epsilon$ in practice. A similar proof also works for the Lipschitz line-search.

---

**Theorem 7** (AdaGrad with Armijo line-search under interpolation). *Under the same assumptions of Proposition 1, but without non-increasing step-sizes, if interpolation is satisfied, AdaGrad with the Armijo line-search and uniform averaging converges at the rate,*

$$\mathbb{E}[f(\bar{w}_T) - f^*] \leq \frac{\left(D^2 + 2\eta_{\max}^2\right)^2 dL_{\max}}{2T} \left(\max\left\{\frac{1}{\eta_{\max}}, \frac{L_{\max}}{a_{\min}}\right\}\right)^2.$$

*where $a_{\min} = \min_k\{\lambda_{\min}(A_k)\}$.*

---

*Proof of Theorem 7.* Following the proof of Proposition 1,

$$2\sum_{k=1}^{T} \eta_k \langle \nabla f_{i_k}(w_k), w_k - w^* \rangle = \sum_{k=1}^{T} \|w_k - w^*\|_{A_k}^2 - \|w_{k+1} - w^*\|_{A_k}^2 + \eta_k^2 \|\nabla f_{i_k}(w_k)\|_{A_k^{-1}}^2.$$

On the left-hand side, we use individual convexity and interpolation, which implies $f_{i_k}(w^*) = \min_w f_{i_k}(w)$ and we can bound $\eta_k$ by $\eta_{\min}$, giving

$$\eta_k \langle \nabla f_{i_k}(w_k), w_k - w^* \rangle \geq \eta_k \underbrace{\left(f_{i_k}(w_k) - f_{i_k}(w^*)\right)}_{\geq 0} \geq \eta_{\min}\left(f_{i_k}(w_k) - f_{i_k}(w^*)\right).$$

On the right-hand side, we can apply the AdaGrad lemmas (Lemma 4)

$$\sum_{k=1}^{T} \|w_k - w^*\|_{A_k}^2 - \|w_{k+1} - w^*\|_{A_k}^2 + \eta_{\max}^2 \|\nabla f_{i_k}(w_k)\|_{A_k^{-1}}^2,$$

$$\leq D^2 \mathsf{Tr}(A_T) + 2\eta_{\max}^2 \mathsf{Tr}(A_T), \qquad\qquad \text{(By Lemmas 3 and 4)}$$

$$\leq \left(D^2 + 2\eta_{\max}^2\right)\sqrt{d}\sqrt{\sum_{k=1}^{T} \|\nabla f_{i_k}(w_k)\|^2}, \qquad \text{(By the trace bound of Lemma 4)}$$

$$\leq \left(D^2 + 2\eta_{\max}^2\right)\sqrt{2dL_{\max}}\sqrt{\sum_{k=1}^{T} f_{i_k}(w_k) - f_{i_k}(w^*)}.$$

$$\text{(By Individual Smoothness and interpolation)}$$

Defining $a = \frac{1}{2\eta_{\min}}\left(D^2 + 2\eta_{\max}^2\right)\sqrt{2dL_{\max}}$ and combining the previous inequalities yields

$$\sum_{k=1}^{T}\left(f_{i_k}(w_k) - f_{i_k}(w^*)\right) \leq a\sqrt{\sum_{k=1}^{T} f_{i_k}(w_k) - f_{i_k}(w^*)}.$$

Taking expectations and applying Jensen's inequality yields

$$\sum_{k=1}^{T} \mathbb{E}[f(w_k) - f(w^*)] \leq a\sqrt{\sum_{k=1}^{T} \mathbb{E}[f(w_k) - f(w^*)]}.$$

Squaring both sides, dividing by $\sum_{k=1}^{T} \mathbb{E}[f(w_k) - f(w^*)]$, followed by dividing by $T$ and applying Jensen's inequality,

$$\mathbb{E}[f(\bar{w}_T) - f(w^*)] \leq \frac{a^2}{T} = \frac{\left(D^2 + 2\eta_{\max}^2\right)^2 dL_{\max}}{2\eta_{\min}^2 T}.$$

Using the Armijo line-search guarantee (Lemma 1) with $c = 1/2$ and a maximum step-size $\eta_{\max}$,

$$\eta_{\min} = \min\left\{\eta_{\max}, \frac{a_{\min}}{L_{\max}}\right\},$$

where $a_{\min} = \min_k\{\lambda_{\min}(A_k)\}$, giving the rate

$$\mathbb{E}[f(\bar{w}_T) - f(w^*)] \leq \frac{\left(D^2 + 2\eta_{\max}^2\right)^2 dL_{\max}}{2T} \left(\max\left\{\frac{1}{\eta_{\max}}, \frac{L_{\max}}{a_{\min}}\right\}\right)^2. \qquad \square$$

## D PROOFS FOR AMSGRAD AND NON-DECREASING PRECONDITIONERS WITHOUT MOMENTUM

We now give the proofs for AMSGrad and general bounded, non-decreasing preconditioners in the smooth setting, using a constant step-size (Theorem 8) and the Armijo line-search (Theorem 4). As in Appendix C, we prove a general proposition and specialize it for each of the theorems;

---

**Proposition 2.** *In addition to assumptions of Theorem 1, assume that (iv) the preconditioners are non-decreasing and have (v) bounded eigenvalues in the $[a_{\min}, a_{\max}]$ range. If the step-sizes are constrained to lie in the range $[\eta_{\min}, \eta_{\max}]$ and satisfy*

$$\eta_k \|\nabla f_{i_k}(w_k)\|_{A_k^{-1}}^2 \leq M(f_{i_k}(w_k) - f_{i_k}^*), \quad \text{for some } M < 2, \tag{7}$$

*using uniform averaging $\bar{w}_T = \frac{1}{T} \sum_{k=1}^{T} w_k$ leads to the rate*

$$\mathbb{E}[f(\bar{w}_T) - f^*] \leq \frac{1}{T} \frac{D^2 d a_{\max}}{(2-M)\eta_{\min}} + \left(\frac{2}{2-M} \frac{\eta_{\max}}{\eta_{\min}} - 1\right)\sigma^2.$$

---

**Theorem 8.** *Under the assumptions of Theorem 1 and assuming (iv) non-decreasing preconditioners (v) bounded eigenvalues in the $[a_{\min}, a_{\max}]$ interval, AMSGrad with no momentum, constant step-size $\eta = \frac{a_{\min}}{2L_{\max}}$ and uniform averaging converges at a rate,*

$$\mathbb{E}[f(\bar{w}_T) - f^*] \leq \frac{2D^2 d\, a_{\max} L_{\max}}{a_{\min} T} + \sigma^2.$$

---

*Proof of Theorem 8.* Using Bounded preconditioner and Individual Smoothness, we have that

$$\|\nabla f_{i_k}(w_k)\|_{A_k^{-1}}^2 \leq \frac{1}{a_{\min}} \|\nabla f_{i_k}(w_k)\|^2 \leq \frac{2L_{\max}}{a_{\min}}(f_{i_k}(w_k) - f_{i_k}^*).$$

A constant step-size $\eta_{\max} = \eta_{\min} = \frac{a_{\min}}{2L_{\max}}$ satisfies the step-size assumption (Eq. 7) with $M = 1$ and

$$\frac{1}{T} \frac{D^2 d a_{\max}}{(2-M)\eta_{\min}} + \left(\frac{2}{2-M} \frac{\eta_{\max}}{\eta_{\min}} - 1\right)\sigma^2 = \frac{1}{T} \frac{2L_{\max} D^2 d a_{\max}}{a_{\min}} + \sigma^2. \qquad \square$$

We restate Theorem 4;

---

**Theorem 4.** *Under the same assumptions as Theorem 1, AMSGrad with zero momentum, Armijo line-search with $c = 3/4$, a step-size upper bound $\eta_{\max}$ and uniform averaging converges at a rate,*

$$\mathbb{E}[f(\bar{w}_T) - f^*] \leq \left(\frac{3D^2 d \cdot a_{\max}}{2T} + 3\eta_{\max}\sigma^2\right) \max\left\{\frac{1}{\eta_{\max}}, \frac{2L_{\max}}{a_{\min}}\right\}.$$

---

*Proof of Theorem 4.* For the Armijo line-search, Lemma 1 guarantees that

$$\eta \|\nabla f_{i_k}(w_k)\|_{A_k^{-1}}^2 \leq \frac{1}{c}(f_{i_k}(w_k) - f_{i_k}^*), \quad \text{and} \quad \min\left\{\eta_{\max}, \frac{2\lambda_{\min}(A_k)(1-c)}{L_{\max}}\right\} \leq \eta \leq \eta_{\max}.$$

Selecting $c = 3/4$ gives $M = 4/3$ and $\eta_{\min} = \min\left\{\eta_{\max}, \frac{a_{\min}}{2L_{\max}}\right\}$, so

$$\frac{1}{T} \frac{D^2 d a_{\max}}{(2-M)\eta_{\min}} + \left(\frac{2}{2-M} \frac{\eta_{\max}}{\eta_{\min}} - 1\right)\sigma^2$$

$$= \frac{1}{T} \frac{D^2 d a_{\max}}{(2-4/3)\eta_{\min}} + \left(\frac{2}{2-4/3} \frac{\eta_{\max}}{\eta_{\min}} - 1\right)\sigma^2,$$

$$= \frac{1}{T} \frac{3D^2 d a_{\max}}{2\eta_{\min}} + \left(\frac{3\eta_{\max}}{\eta_{\min}} - 1\right)\sigma^2,$$

$$\leq \frac{3D^2 d a_{\max}}{2T} \max\left\{\frac{1}{\eta_{\max}}, \frac{2L_{\max}}{a_{\min}}\right\} + 3\eta_{\max}\sigma^2 \max\left\{\frac{1}{\eta_{\max}}, \frac{2L_{\max}}{a_{\min}}\right\}. \qquad \square$$

**Theorem 9.** *Under the assumptions of Theorem 1 and assuming (iv) non-decreasing precondition-ers (v) bounded eigenvalues in the $[a_{\min}, a_{\max}]$ interval, AMSGrad with no momentum, Armijo SPS with $c = 3/4$ and uniform averaging converges at a rate,*

$$\mathbb{E}[f(\bar{w}_T) - f^*] \leq \left(\frac{3D^2 d \cdot a_{\max}}{2T} + 3\eta_{\max}\sigma^2\right) \max\left\{\frac{1}{\eta_{\max}}, \frac{3L_{\max}}{2a_{\min}}\right\}.$$

*Proof of Theorem 5.* For Armijo SPS, Lemma 2 guarantees that

$$\eta_k \|\nabla f_{i_k}(w_k)\|^2_{A_k^{-1}} \leq \frac{1}{c}(f_{i_k}(w_k) - f_{i_k}^*), \quad \text{and} \quad \min\left\{\eta_{\max}, \frac{a_{\min}}{2c L_{\max}}\right\} \leq \eta \leq \eta_{\max}.$$

Selecting $c = 3/4$ gives $M = 4/3$ and $\eta_{\min} = \min\left\{\eta_{\max}, \frac{2a_{\min}}{3L_{\max}}\right\}$, so

$$\frac{1}{T}\frac{D^2 d a_{\max}}{(2-M)\eta_{\min}} + \left(\frac{2}{2-M}\frac{\eta_{\max}}{\eta_{\min}} - 1\right)\sigma^2$$
$$= \frac{1}{T}\frac{D^2 d a_{\max}}{(2-4/3)\eta_{\min}} + \left(\frac{2}{2-4/3}\frac{\eta_{\max}}{\eta_{\min}} - 1\right)\sigma^2,$$
$$= \frac{1}{T}\frac{3D^2 d a_{\max}}{2\eta_{\min}} + \left(\frac{3\eta_{\max}}{\eta_{\min}} - 1\right)\sigma^2,$$
$$\leq \frac{3D^2 d a_{\max}}{2T} \max\left\{\frac{1}{\eta_{\max}}, \frac{3L_{\max}}{2a_{\min}}\right\} + 3\eta_{\max}\sigma^2 \max\left\{\frac{1}{\eta_{\max}}, \frac{3L_{\max}}{2a_{\min}}\right\}. \qquad \square$$

Before diving into the proof of Proposition 2, we prove the following lemma to handle terms of the form $\eta_k(f_{i_k}(w_k) - f_{i_k}(w^*))$. If $\eta_k$ depends on the function sampled at the current iteration, $f_{i_k}$, as in the case of line-search, we cannot take expectations as the terms are not independent. Lemma 6 bounds $\eta_k(f_{i_k}(w_k) - f_{i_k}(w^*))$ in terms of the range $[\eta_{\min}, \eta_{\max}]$;

**Lemma 6.** *If $0 \leq \eta_{\min} \leq \eta \leq \eta_{\max}$ and the minimum value of $f_i$ is $f_i^*$, then*

$$\eta(f_i(w) - f_i(w^*)) \geq \eta_{\min}(f_i(w) - f_i(w^*)) - (\eta_{\max} - \eta_{\min})(f_i(w^*) - f_i^*).$$

*Proof of Lemma 6.* By adding and subtracting $f_i^*$, the minimum value of $f_i$, we get a non-negative and a non-positive term multiplied by $\eta$. We can use the bounds $\eta \geq \eta_{\min}$ and $\eta \leq \eta_{\max}$ separately;

$$\eta[f_i(w) - f_i(w^*)] = \eta[\underbrace{f_i(w) - f_i^*}_{\geq 0} + \underbrace{f_i^* - f_i(w^*)}_{\leq 0}],$$
$$\geq \eta_{\min}[f_i(w) - f_i^*] + \eta_{\max}[f_i^* - f_i(w^*)].$$

Adding and subtracting $\eta_{\min}f_i(w^*)$ finishes the proof,

$$= \eta_{\min}[f_i(w) - f_i(w^*) + f_i(w^*) - f_i^*] + \eta_{\max}[f_i^* - f_i(w^*)],$$
$$= \eta_{\min}[f_i(w) - f_i(w^*)] + (\eta_{\max} - \eta_{\min})[f_i^* - f_i(w^*)]. \qquad \square$$

*Proof of Proposition 2.* We start with the Update rule, measuring distances to $w^*$ in the $\|\cdot\|_{A_k}$ norm,

$$\|w_{k+1} - w^*\|^2_{A_k} = \|w_k - w^*\|^2_{A_k} - 2\eta_k\langle\nabla f_{i_k}(w_k), w_k - w^*\rangle + \eta_k^2\|\nabla f_{i_k}(w_k)\|^2_{A_k^{-1}} \quad (8)$$

To bound the RHS, we use the assumption on the step-sizes (Eq. (7)) and Individual Convexity,

$$-2\eta_k\langle\nabla f_{i_k}(w_k), w_k - w^*\rangle + \eta_k^2\|\nabla f_{i_k}(w_k)\|^2_{A_k^{-1}},$$
$$\leq -2\eta_k\langle\nabla f_{i_k}(w_k), w_k - w^*\rangle + M\eta_k(f_{i_k}(w_k) - f_{i_k}^*), \qquad \text{(Step-size assumption, Eq. (7))}$$
$$\leq -2\eta_k[f_{i_k}(w_k) - f_{i_k}(w^*)] + M\eta_k(f_{i_k}(w_k) - f_{i_k}^*), \qquad \text{(Individual Convexity)}$$
$$\leq -2\eta_k[f_{i_k}(w_k) - f_{i_k}(w^*)] + M\eta_k(f_{i_k}(w_k) - f_{i_k}(w^*) + f_{i_k}(w^*) - f_{i_k}^*), \qquad (\pm f_{i_k}(w^*))$$
$$\leq -(2-M)\eta_k[f_{i_k}(w_k) - f_{i_k}(w^*)] + M\eta_{\max}(f_{i_k}(w^*) - f_{i_k}^*). \qquad (\eta_k \leq \eta_{\max})$$

Plugging the inequality back into Eq. (8) and reorganizing the terms yields

$$(2 - M)\eta_k[f_{i_k}(w_k) - f_{i_k}(w^*)] \leq \left( \|w_k - w^*\|^2_{A_k} - \|w_{k+1} - w^*\|^2_{A_k} \right)$$
$$+ M\eta_{\max}(f_{i_k}(w^*) - f^*_{i_k}) \tag{9}$$

Using Lemma 6, we have that

$$(2 - M)\eta_k[f_{i_k}(w_k) - f_{i_k}(w^*)] \geq (2 - M)\eta_{\min}(f_{i_k}(w_k) - f_{i_k}(w^*))$$
$$- (2 - M)(\eta_{\max} - \eta_{\min})(f_{i_k}(w^*) - f^*_{i_k}).$$

Using this inequality in Eq. (9), we have that

$$(2 - M)\eta_{\min}(f_{i_k}(w_k) - f_{i_k}(w^*)) - (2 - M)(\eta_{\max} - \eta_{\min})(f_{i_k}(w^*) - f^*_{i_k})$$
$$\leq \left( \|w_k - w^*\|^2_{A_k} - \|w_{k+1} - w^*\|^2_{A_k} \right) + M\eta_{\max}(f_{i_k}(w^*) - f^*_{i_k}),$$

Moving the terms depending on $f_{i_k}(w^*) - f^*_{i_k}$ to the RHS,

$$(2 - M)\eta_{\min}(f_{i_k}(w_k) - f_{i_k}(w^*)) \leq \left( \|w_k - w^*\|^2_{A_k} - \|w_{k+1} - w^*\|^2_{A_k} \right)$$
$$+ (2\eta_{\max} - (2 - M)\eta_{\min})(f_{i_k}(w^*) - f^*_{i_k}).$$

Taking expectations and summing across iterations yields

$$(2 - M)\eta_{\min} \sum_{k=1}^{T} \mathbb{E}[f_{i_k}(w_k) - f_{i_k}(w^*)] \leq \mathbb{E}\left[ \sum_{k=1}^{T} \left( \|w_k - w^*\|^2_{A_k} - \|w_{k+1} - w^*\|^2_{A_k} \right) \right]$$
$$+ (2\eta_{\max} - (2 - M)\eta_{\min})T\sigma^2.$$

Using Lemma 3 to telescope the distances and using the Bounded preconditioner,

$$\sum_{k=1}^{T} \|w_k - w^*\|^2_{A_k} - \|w_{k+1} - w^*\|^2_{A_k} \leq \sum_{k=1}^{T} \|w_k - w^*\|^2_{A_k - A_{k-1}} \leq D^2 \, \mathsf{Tr}(A_T) \leq D^2 \, d \, a_{\max},$$

which guarantees that

$$(2 - M)\eta_{\min} \sum_{k=1}^{T} \mathbb{E}[f(w_k) - f(w^*)] \leq D^2 d a_{\max} + (2\eta_{\max} - (2 - M)\eta_{\min})T\sigma^2.$$

Dividing by $T(2 - M)\eta_{\min}$ and using Jensen's inequality finishes the proof, giving the rate for the averaged iterate,

$$\mathbb{E}[f(\bar{w}_T) - f(w^*)] \leq \frac{1}{T} \frac{D^2 d a_{\max}}{(2 - M)\eta_{\min}} + \left( \frac{2}{2 - M} \frac{\eta_{\max}}{\eta_{\min}} - 1 \right)\sigma^2. \qquad \square$$

# E  AMSGRAD WITH MOMENTUM

We first show the relation between the AMSGrad momentum and heavy ball momentum and then present the proofs with AMSGrad momentum in E.2 and heavy ball momentum in E.3.

## E.1  RELATION BETWEEN THE AMSGRAD UPDATE AND PRECONDITIONED SGD WITH HEAVY-BALL MOMENTUM

Recall that the AMSGrad update is given as:

$$w_{k+1} = w_k - \eta_k A_k^{-1} m_k \quad ; \quad m_k = \beta m_{k-1} + (1 - \beta)\nabla f_{i_k}(w_k)$$

Simplifying,

$$w_{k+1} = w_k - \eta_k A_k^{-1}(\beta m_{k-1} + (1 - \beta)\nabla f_{i_k}(w_k))$$
$$w_{k+1} = w_k - \eta_k(1 - \beta) A_k^{-1}\nabla f_{i_k}(w_k) - \eta_k\beta A_k^{-1} m_{k-1}$$

From the update at iteration $k - 1$,

$$w_k = w_{k-1} - \eta_{k-1} A_{k-1}^{-1} m_{k-1}$$
$$\implies -m_{k-1} = \frac{1}{\eta_{k-1}} A_{k-1}(w_k - w_{k-1})$$

From the above relations,

$$w_{k+1} = w_k - \eta_k(1 - \beta) A_k^{-1}\nabla f_{i_k}(w_k) + \beta\,\frac{\eta_k}{\eta_{k-1}}\,A_k^{-1} A_{k-1}(w_k - w_{k-1})$$

which is of the same form as

$$w_{k+1} = w_k - \eta_k A_k^{-1} + \gamma(w_k - w_{k-1}),$$

the update with heavy ball momentum. The two updates are equivalent up to constants except for the key difference that for AMSGrad, the momentum vector $(w_k - w_{k-1})$ is further preconditioned by $A_k^{-1} A_{k-1}$.

### E.2 PROOFS FOR AMSGRAD WITH MOMENTUM

We now give the proofs for AMSGrad having the update.

$$w_{k+1} = w_k - \eta_k A_k^{-1} m_k \quad ; \quad m_k = \beta m_{k-1} + (1 - \beta) \nabla f_{i_k}(w_k)$$

We analyze it in the smooth setting using a constant step-size (Theorem 3), conservative Armijo SPS (Theorem 5) and conservative Armijo SLS (Theorem 10). As before, we abstract the common elements to a general proposition and specialize it for each of the theorems.

---

**Proposition 3.** *In addition to assumptions of Theorem 1, assume that (iv) the preconditioners are non-decreasing and have (v) bounded eigenvalues in the $[a_{\min}, a_{\max}]$ range. If the step-sizes are lower-bounded and non-increasing, $\eta_{\min} \leq \eta_k \leq \eta_{k-1}$ and satisfy*

$$\eta_k \left\| \nabla f_{i_k}(w_k) \right\|_{A_k^{-1}}^2 \leq M(f_{i_k}(w_k) - f_{i_k}^*), \quad \text{for some } M < 2\frac{1-\beta}{1+\beta}, \tag{10}$$

*using uniform averaging $\bar{w}_T = \frac{1}{T} \sum_{k=1}^{T} w_k$ leads to the rate*

$$\mathbb{E}[f(\bar{w}_T) - f^*] \leq \frac{1+\beta}{1-\beta}\left(2 - \frac{1+\beta}{1-\beta}M\right)^{-1}\left[\frac{D^2 d a_{\max}}{\eta_{\min} T} + M\sigma^2\right].$$

---

We first show how the convergence rate of each step-size method can be derived from Proposition 3.

---

**Theorem 3.** *Under the same assumptions as Theorem 1, and assuming (iv) non-decreasing preconditioners (v) bounded eigenvalues in the $[a_{\min}, a_{\max}]$ interval, where $\kappa = a_{\max}/a_{\min}$, AMSGrad with $\beta \in [0, 1)$, constant step-size $\eta = \frac{1-\beta}{1+\beta}\frac{a_{\min}}{2L_{\max}}$ and uniform averaging converges at a rate,*

$$\mathbb{E}[f(\bar{w}_T) - f^*] \leq \left(\frac{1+\beta}{1-\beta}\right)^2 \frac{2L_{\max}D^2 d\kappa}{T} + \sigma^2.$$

---

*Proof of Theorem 3.* Using Bounded preconditioner and Individual Smoothness, we have that

$$\eta \left\| \nabla f_{i_k}(w_k) \right\|_{A_k^{-1}}^2 \leq \eta \frac{1}{a_{\min}} \left\| \nabla f_{i_k}(w_k) \right\|^2 \leq \eta \frac{2L_{\max}}{a_{\min}}(f_{i_k}(w_k) - f_{i_k}^*).$$

Using a constant step-size $\eta = \frac{1-\beta}{1+\beta}\frac{a_{\min}}{2L_{\max}}$ satisfies the requirement of Proposition 3 (Eq. (10)) with constant $M = \frac{1-\beta}{1+\beta}$. The convergence is then,

$$\begin{aligned}
\mathbb{E}[f(\bar{w}_T) - f(w^*)] &\leq \frac{1+\beta}{1-\beta}\left(2 - \frac{1+\beta}{1-\beta}M\right)^{-1}\left[\frac{D^2 d a_{\max}}{\eta_{\min} T} + M\sigma^2,\right] \\
&= \frac{1+\beta}{1-\beta}\left[\frac{D^2 d a_{\max}}{\frac{1-\beta}{1+\beta}\frac{a_{\min}}{2L_{\max}}T} + \frac{1-\beta}{1+\beta}\sigma^2,\right] \\
&= \left(\frac{1+\beta}{1-\beta}\right)^2 \frac{2L_{\max}D^2 d\kappa}{T} + \sigma^2,
\end{aligned}$$

with $\kappa = a_{\max}/a_{\min}$.

$\square$

**Theorem 5.** *Under the same assumptions of Theorem 1 and assuming (iv) non-decreasing preconditioners (v) bounded eigenvalues in the $[a_{\min}, a_{\max}]$ interval with $\kappa = a_{\max}/a_{\min}$, AMSGrad with $\beta \in [0, 1)$, conservative Armijo SPS with $c = 1+\beta/1-\beta$ and uniform averaging converges at a rate,*

$$\mathbb{E}[f(\bar{w}_T) - f^*] \leq \left(\frac{1+\beta}{1-\beta}\right)^2 \frac{2L_{\max}D^2 d\kappa}{T} + \sigma^2.$$

*Proof of Theorem 5.* For Armijo SPS, Lemma 2 guarantees that

$$\eta_k \|\nabla f_{i_k}(w_k)\|_{A_k^{-1}}^2 \leq \frac{1}{c}(f_{i_k}(w_k) - f_{i_k}^*), \qquad \text{and} \qquad \frac{a_{\min}}{2c\,L_{\max}} \leq \eta_k.$$

Setting $c = \frac{1+\beta}{1-\beta}$ ensures that $M = 1/c$ satisfies the requirement of Proposition 3 and $\eta_{\min} \geq \frac{1-\beta}{1+\beta}\frac{a_{\min}}{2L_{\max}}$. Plugging in these values into Proposition 3 completes the proof. □

**Theorem 10.** *Under the assumptions of Theorem 1 and assuming (iv) non-decreasing preconditioners (v) bounded eigenvalues in the $[a_{\min}, a_{\max}]$ interval, AMSGrad with momentum with parameter $\beta \in [0, 1/5)$, conservative Armijo SLS with $c = \frac{2}{3}\frac{1+\beta}{1-\beta}$ and uniform averaging converges at a rate,*

$$\mathbb{E}[f(\bar{w}_T) - f^*] \leq 3\frac{1+\beta}{1-5\beta}\frac{L_{\max}D^2 d\kappa}{T} + 3\sigma^2$$

*Proof of Theorem 10.* For Armijo SLS, Lemma 1 guarantees that

$$\eta_k \|\nabla f_{i_k}(w_k)\|_{A_k^{-1}}^2 \leq \frac{1}{c}(f_{i_k}(w_k) - f_{i_k}^*), \qquad \text{and} \qquad \frac{2(1-c)\,a_{\min}}{L_{\max}} \leq \eta_k.$$

The line-search parameter $c$ is restricted to $[0, 1]$ and relates to the the requirement parameter $M$ of Proposition 3 (Eq. (10)) through $M = 1/c$. The combined requirements on $M$ are then that $1 < M < 2\frac{1-\beta}{1+\beta}$, which is only feasible if $\beta < \frac{1}{3}$. To leave room to satisfy the constraints, let $\beta < \frac{1}{5}$.

Setting $\frac{1}{c} = M = \frac{3}{2}\frac{1-\beta}{1+\beta}$ satisfies the constraints and requirement for Proposition 3, and

$$
\begin{aligned}
\mathbb{E}[f(\bar{w}_T) - f(w^*)] &\leq \frac{1+\beta}{1-\beta}\left(2 - \frac{1+\beta}{1-\beta}M\right)^{-1}\left[\frac{D^2 d a_{\max}}{\eta_{\min}T} + M\sigma^2\right], \\
&= \frac{1+\beta}{1-\beta}\left(2 - \frac{3}{2}\right)^{-1}\left[\frac{L_{\max}}{2(1-c)\,a_{\min}}\frac{D^2 d a_{\max}}{T} + \frac{3}{2}\frac{1-\beta}{1+\beta}\sigma^2\right], \\
&= \frac{1+\beta}{1-\beta}\frac{L_{\max}}{(1-c)}\frac{D^2 d\kappa}{T} + 3\sigma^2 = 3\frac{1+\beta}{1-5\beta}\frac{L_{\max}D^2 d\kappa}{T} + 3\sigma^2.
\end{aligned}
$$

where the last step substituted $1/(1-c)$,

$$1 - c = 1 - \frac{2}{3}\frac{1+\beta}{1-\beta} = \frac{3(1-\beta) - 2(1+\beta)}{3(1-\beta)} = \frac{1}{3}\frac{1-5\beta}{1-\beta}. \qquad \square$$

Before diving into the proof of Proposition 3, we prove the following lemma,

**Lemma 7.** *For any set of vectors $a, b, c, d$, if $a = b + c$, then,*

$$\|a - d\|^2 = \|b - d\|^2 - \|a - b\|^2 + 2\langle c, a - d\rangle$$

*Proof.*

$$\|a - d\|^2 = \|b + c - d\|^2 = \|b - d\|^2 + 2\langle c, b - d\rangle + \|c\|^2$$

Since $c = a - b$,

$$
\begin{aligned}
&= \|b - d\|^2 + 2\langle a - b, b - d\rangle + \|a - b\|^2 \\
&= \|b - d\|^2 + 2\langle a - b, b - a + a - d\rangle + \|a - b\|^2 \\
&= \|b - d\|^2 + 2\langle a - b, b - a\rangle + 2\langle a - b, a - d\rangle + \|a - b\|^2 \\
&= \|b - d\|^2 - 2\|a - b\|^2 + 2\langle a - b, a - d\rangle + \|a - b\|^2 \\
&= \|b - d\|^2 - \|a - b\|^2 + 2\langle c, a - d\rangle
\end{aligned}
$$

$\square$

We now move to the proof of the main proposition. Our proof follows the structure of Reddi et al. (2018); Alacaoglu et al. (2020).

*Proof of Proposition 3.* To reduce clutter, let $P_k = A_k/\eta_k$. Using the update, we have the expansion

$$
\begin{aligned}
w_{k+1} - w^* &= \left(w_k - P_k^{-1} m_k\right) - w^*, \\
&= \left(w_k - (1 - \beta) P_k^{-1} \nabla f_{i_k}(w_k) - \beta P_k^{-1} m_{k-1}\right) - w^*,
\end{aligned}
$$

Measuring distances in the $\|\cdot\|_{P_k}$-norm, such that $\|x\|_{P_k}^2 = \langle x, P_k x\rangle$,

$$
\begin{aligned}
\|w_{k+1} - w^*\|_{P_k}^2 = \|w_k - w^*\|_{P_k}^2 &- 2(1 - \beta)\langle w_k - w^*, \nabla f_{i_k}(w_k)\rangle, \\
&- 2\beta\langle w_k - w^*, m_{k-1}\rangle + \|m_k\|_{P_k^{-1}}^2 .
\end{aligned}
$$

We separate the distance to $w^*$ from the momentum in the second inner product using the update and Lemma 7 with $a = c = P_{k-1}^{1/2}(w_k - w^*)$, $b = \mathbf{0}$, $d = P_{k-1}^{1/2}(w_{k-1} - w^*)$.

$$
\begin{aligned}
-2\langle m_{k-1}, w_k - w^*\rangle &= -2\langle P_{k-1}(w_{k-1} - w_k), w_k - w^*\rangle, \\
&= \left[\|w_k - w_{k-1}\|_{P_{k-1}}^2 + \|w_k - w^*\|_{P_{k-1}}^2 - \|w_{k-1} - w^*\|_{P_{k-1}}^2\right], \\
&= \|m_{k-1}\|_{P_{k-1}^{-1}}^2 + \|w_k - w^*\|_{P_{k-1}}^2 - \|w_{k-1} - w^*\|_{P_{k-1}}^2, \\
&\leq \|m_{k-1}\|_{P_{k-1}^{-1}}^2 + \|w_k - w^*\|_{P_k}^2 - \|w_{k-1} - w^*\|_{P_{k-1}}^2,
\end{aligned}
$$

where the last inequality uses the fact that $\eta_k \leq \eta_{k-1}$ and $A_k \succeq A_{k-1}$, which implies $P_k \succeq P_{k-1}$, and $\|w_k - w^*\|_{P_{k-1}}^2 \leq \|w_k - w^*\|_{P_k}^2$. Plugging this inequality in and grouping terms yields

$$
\begin{aligned}
2(1 - \beta)\langle w_k - w^*, \nabla f_{i_k}(w_k)\rangle &\leq \left[\|w_k - w^*\|_{P_k}^2 - \|w_{k+1} - w^*\|_{P_k}^2\right] \\
&+ \beta\left[\|w_k - w^*\|_{P_k}^2 - \|w_{k-1} - w^*\|_{P_{k-1}}^2\right] \\
&+ \left[\beta\|m_{k-1}\|_{P_{k-1}^{-1}}^2 + \|m_k\|_{P_k^{-1}}^2\right]
\end{aligned}
$$

By convexity, the inner product on the left-hand-side is bounded by $\langle w_k - w^*, \nabla f_{i_k}(w_k)\rangle \geq f_{i_k}(w_k) - f_{i_k}(w^*)$. The first two lines of the right-hand-side will telescope if we sum all iterations, so we only need to treat the norms of the momentum terms. We introduce a free parameter $\delta \geq 0$, that is only used for the analysis, and expand

$$\beta\|m_{k-1}\|_{P_{k-1}^{-1}}^2 + \|m_k\|_{P_k^{-1}}^2 = \beta\|m_{k-1}\|_{P_{k-1}^{-1}}^2 + (1 + \delta)\|m_k\|_{P_k^{-1}}^2 - \delta\|m_k\|_{P_k^{-1}}^2 .$$

To bound $\|m_k\|_{P_k^{-1}}^2$, we expand it by its update and use Young's inequality to get

$$
\begin{aligned}
\|m_k\|_{P_k^{-1}}^2 &= \|\beta m_{k-1} + (1 - \beta)\nabla f_{i_k}(w_k)\|_{P_k^{-1}}^2 \\
&\leq (1 + \epsilon)\beta^2 \|m_{k-1}\|_{P_k^{-1}}^2 + (1 + 1/\epsilon)(1 - \beta)^2 \|\nabla f_{i_k}(w_k)\|_{P_k^{-1}}^2,
\end{aligned}
$$

where $\epsilon > 0$ is also a free parameter, introduced to control the tradeoff of the bound. Plugging this bound in the momentum terms, we get

$$\beta \left\| m_{k-1} \right\|_{P_{k-1}^{-1}}^2 + \left\| m_k \right\|_{P_k^{-1}}^2 \leq \beta \left\| m_{k-1} \right\|_{P_{k-1}^{-1}}^2 + (1+\epsilon)(1+\delta)\beta^2 \left\| m_{k-1} \right\|_{P_k^{-1}}^2 - \delta \left\| m_k \right\|_{P_k^{-1}}^2 ,$$
$$+ (1 + 1/\epsilon)(1+\delta)(1-\beta)^2 \left\| \nabla f_{i_k}(w_k) \right\|_{P_k^{-1}}^2 .$$

As $P_k^{-1} \preceq P_{k-1}^{-1}$, we have that $\left\| m_{k-1} \right\|_{P_k^{-1}}^2 \leq \left\| m_{k-1} \right\|_{P_{k-1}^{-1}}^2$ which implies

$$\leq \left( \beta + (1+\epsilon)(1+\delta)\beta^2 \right) \left\| m_{k-1} \right\|_{P_{k-1}^{-1}}^2 - \delta \left\| m_k \right\|_{P_k^{-1}}^2$$
$$+ (1 + 1/\epsilon)(1+\delta)(1-\beta)^2 \left\| \nabla f_{i_k}(w_k) \right\|_{P_k^{-1}}^2 .$$

To get a telescoping sum, we set $\delta$ to be equal to $\beta + (1+\epsilon)(1+\delta)\beta^2$, which is satisfied if $\delta = \frac{\beta + (1+\epsilon)\beta^2}{1 - (1+\epsilon)\beta^2}$, and $\delta > 0$ is satisfied if $\beta < 1/\sqrt{1+\epsilon}$. We now plug back the inequality

$$\beta \left\| m_{k-1} \right\|_{P_{k-1}^{-1}}^2 + \left\| m_k \right\|_{P_k^{-1}}^2 \leq \delta \left[ \left\| m_{k-1} \right\|_{P_{k-1}^{-1}}^2 - \left\| m_k \right\|_{P_k^{-1}}^2 \right]$$
$$+ (1 + 1/\epsilon)(1+\delta)(1-\beta)^2 \left\| \nabla f_{i_k}(w_k) \right\|_{P_k^{-1}}^2 ,$$

in the previous expression to get

$$2(1-\beta)\left( f_{i_k}(w_k) - f_{i_k}(w^*) \right) \leq \left\| w_k - w^* \right\|_{P_k}^2 - \left\| w_{k+1} - w^* \right\|_{P_k}^2$$
$$+ \beta \left[ \left\| w_k - w^* \right\|_{P_k}^2 - \left\| w_{k-1} - w^* \right\|_{P_{k-1}}^2 \right]$$
$$+ \delta \left[ \left\| m_{k-1} \right\|_{P_{k-1}^{-1}}^2 - \left\| m_k \right\|_{P_k^{-1}}^2 \right]$$
$$+ (1 + 1/\epsilon)(1+\delta)(1-\beta)^2 \left\| \nabla f_{i_k}(w_k) \right\|_{P_k^{-1}}^2 .$$

All terms now telescope, except the gradient norm which we bound using the step size assumption,

$$\left\| \nabla f_{i_k}(w_k) \right\|_{P_k^{-1}}^2 = \eta_k \left\| \nabla f_{i_k}(w_k) \right\|_{A_k^{-1}}^2 \leq M(f_{i_k}(w_k) - f_{i_k}^*),$$
$$= M(f_{i_k}(w_k) - f_{i_k}(w^*)) + M(f_{i_k}(w^*) - f_{i_k}^*).$$

This gives the expression

$$\alpha \left( f_{i_k}(w_k) - f_{i_k}(w^*) \right) \leq \left\| w_k - w^* \right\|_{P_k}^2 - \left\| w_{k+1} - w^* \right\|_{P_k}^2$$
$$+ \beta \left[ \left\| w_k - w^* \right\|_{P_k}^2 - \left\| w_{k-1} - w^* \right\|_{P_{k-1}}^2 \right]$$
$$+ \delta \left[ \left\| m_{k-1} \right\|_{P_{k-1}^{-1}}^2 - \left\| m_k \right\|_{P_k^{-1}}^2 \right]$$
$$+ (1 + 1/\epsilon)(1+\delta)(1-\beta)^2 M(f_{i_k}(w^*) - f_{i_k}^*),$$

with $\alpha = 2(1-\beta) - (1 + 1/\epsilon)(1+\delta)(1-\beta)^2 M$. Summing all iterations, the individual terms are bounded by the Bounded iterates and Lemma 3;

$$\sum_{k=1}^T \left\| w_k - w^* \right\|_{P_k}^2 - \left\| w_{k+1} - w^* \right\|_{P_k}^2 \qquad \leq D^2 \mathsf{Tr}(P_T) \qquad \leq \frac{D^2}{\eta_{\min}} \mathsf{Tr}(A_T)$$

$$\beta \sum_{k=1}^T \left\| w_k - w^* \right\|_{P_k}^2 - \left\| w_{k-1} - w^* \right\|_{P_{k-1}}^2 \qquad \leq \beta \left\| w_T - w^* \right\|_{P_T}^2 \qquad \leq \beta \frac{D^2}{\eta_{\min}} \mathsf{Tr}(A_T)$$

$$\delta \sum_{k=1}^T \left\| m_{k-1} \right\|_{P_{k-1}^{-1}}^2 - \left\| m_k \right\|_{P_k^{-1}}^2 \qquad \leq \delta \left\| m_0 \right\|_{P_0}^2 \qquad = 0.$$

Using the boundedness of the preconditioners gives $\mathsf{Tr}(A_T) \leq d a_{\max}$ and the total bound

$$\alpha \sum_{k=1}^T (f_{i_k}(w_k) - f_{i_k}(w^*)) \leq \frac{(1+\beta)D^2 d a_{\max}}{\eta_{\min}} + (1 + 1/\epsilon)(1+\delta)(1-\beta)^2 M \sum_{k=1}^T (f_{i_k}(w^*) - f_{i_k}^*).$$

Taking expectations,

$$\alpha \sum_{k=1}^{T} \mathbb{E}[f(w_k) - f(w^*)] \leq \frac{(1+\beta)D^2 da_{\max}}{\eta_{\min}} + (1 + 1/\epsilon)(1+\delta)(1-\beta)^2 M\sigma^2 T.$$

It remains to expand $\alpha$ and simplify the constants. We had defined

$$\alpha = 2(1-\beta) - (1 + 1/\epsilon)(1+\delta)(1-\beta)^2 M > 0, \qquad \text{and} \qquad \delta = \frac{\beta + (1+\epsilon)\beta^2}{1 - (1+\epsilon)\beta^2} > 0,$$

where $\epsilon > 0$ is a free parameter. This puts the requirement on $\beta$ that $\beta < 1/\sqrt{1+\epsilon}$. To simplify the bounds, we set $\beta = 1/(1+\epsilon)$, $\epsilon = 1/\beta - 1$, which gives the substitutions

$$1 + \epsilon = \frac{1}{\beta} \qquad 1 + \frac{1}{\epsilon} = \frac{1}{1-\beta} \qquad \delta = 2\frac{\beta}{1-\beta} \qquad 1 + \delta = \frac{1+\beta}{1-\beta}.$$

Plugging those into the rate gives

$$\alpha \sum_{k=1}^{T} \mathbb{E}[f(w_k) - f(w^*)] \leq \frac{(1+\beta)D^2 da_{\max}}{\eta_{\min}} + (1+\beta)M\sigma^2 T,$$

while plugging them into $\alpha$ gives

$$\alpha = 2(1-\beta) - (1 + 1/\epsilon)(1+\delta)(1-\beta)^2 M,$$
$$= (1-\beta)\left[2 - \frac{1+\beta}{1-\beta}M\right], \text{ which is positive if } M < 2\frac{1-\beta}{1+\beta}.$$

Dividing by $\alpha T$, using Jensen's inequality and averaging finishes the proof, with the rate

$$\sum_{k=1}^{T} \mathbb{E}[f(w_k) - f(w^*)] \leq \frac{1+\beta}{1-\beta}\left(2 - \frac{1+\beta}{1-\beta}M\right)^{-1}\left[\frac{D^2 da_{\max}}{\eta_{\min}T} + M\sigma^2\right]. \qquad \square$$

### E.3 Proofs for AMSGrad with heavy ball momentum

We now give the proofs for AMSGrad with heavy ball momentum with the update.

$$w_{k+1} = w_k - \eta_k A_k^{-1} \nabla f_{i_k}(w_k) + \gamma (w_k - w_{k-1})$$

We analyze it in the smooth setting using a constant step-size (Theorem 11), a conservative Armijo SPS (Theorem 12) and conservative Armijo SLS (Theorem 13). As before, we abstract the common elements to a general proposition and specialize it for each of the theorems.

---

**Proposition 4.** *In addition to assumptions of Theorem 1, assume that (iv) the preconditioners are non-decreasing and have (v) bounded eigenvalues in the $[a_{\min}, a_{\max}]$ range. If the step-sizes are lower-bounded and non-increasing, $\eta_{\min} \leq \eta_k \leq \eta_{k-1}$ and satisfy*

$$\eta_k \|\nabla f_{i_k}(w_k)\|^2_{A_k^{-1}} \leq M(f_{i_k}(w_k) - f_{i_k}^*), \quad \text{for some } M < 2 - 2\gamma, \tag{11}$$

*AMSGrad with heavy ball momentum with parameter $\gamma < 1$ and uniform averaging $\bar{w}_T = \frac{1}{T}\sum_{k=1}^{T} w_k$ leads to the rate*

$$\mathbb{E}[f(\bar{w}_T) - f^*] \leq \frac{1}{2-2\gamma-M}\left[\frac{1}{T}\left(\frac{2(1+\gamma^2)D^2 a_{\max}d}{\eta_{\min}} + 2\gamma[f(w_0) - f(w^*)]\right) + M\sigma^2\right].$$

---

We first show how the convergence rate of each step-size method can be derived from Proposition 4.

---

**Theorem 11.** *Under the assumptions of Theorem 1 and assuming (iv) non-decreasing preconditioners (v) bounded eigenvalues in the $[a_{\min}, a_{\max}]$ range, AMSGrad with heavy ball momentum with parameter $\gamma \in [0, 1)$, constant step-size $\eta = \frac{2a_{\min}(1-\gamma)}{3L_{\max}}$ and uniform averaging converges at a rate*

$$\mathbb{E}[f(\bar{w}_T) - f^*] \leq \frac{1}{T}\left(\frac{9}{2}\frac{1+\gamma^2}{(1-\gamma)^2}L_{\max}D^2\kappa d + \frac{3\gamma}{(1-\gamma)}[f(w_0) - f(w^*)]\right) + 2\sigma^2.$$

---

*Proof of Theorem 11.* Using Bounded preconditioner and Individual Smoothness, we have that

$$\eta \|\nabla f_{i_k}(w_k)\|^2_{A_k^{-1}} \leq \eta \frac{1}{a_{\min}}\|\nabla f_{i_k}(w_k)\|^2 \leq \eta\frac{2L_{\max}}{a_{\min}}(f_{i_k}(w_k) - f_{i_k}^*).$$

A constant step-size $\eta = 2a_{\min}(1-\gamma)/3L_{\max}$ means the requirement for Proposition 4 is satisfied with $M = \frac{4}{3}(1-\gamma)$. Plugging $(2 - 2\gamma - M) = \frac{2}{3}(1-\gamma)$ in Proposition 4 finishes the proof. $\square$

---

**Theorem 12.** *Under the assumptions of Theorem 1 and assuming (iv) non-decreasing preconditioners (v) bounded eigenvalues in the $[a_{\min}, a_{\max}]$ interval, AMSGrad with heavy ball momentum with parameter $\gamma \in [0, 1)$, conservative Armijo SPS with $c = 3/4(1-\gamma)$ and uniform averaging converges at a rate,*

$$\mathbb{E}[f(\bar{w}_T) - f^*] \leq \frac{1}{T}\left(\frac{9}{2}\frac{1+\gamma^2}{(1-\gamma)^2}L_{\max}D^2\kappa d + \frac{3\gamma}{(1-\gamma)}[f(w_0) - f(w^*)]\right) + 2\sigma^2.$$

---

*Proof of Theorem 12.* For Armijo SPS, Lemma 2 guarantees that

$$\eta_k \|\nabla f_{i_k}(w_k)\|^2_{A_k^{-1}} \leq \frac{1}{c}(f_{i_k}(w_k) - f_{i_k}^*), \qquad \text{and} \qquad \frac{a_{\min}}{2c\,L_{\max}} \leq \eta_k.$$

Selecting $c = 3/4(1-\gamma)$ gives $M = 4/3(1-\gamma) \leq 2(1-\gamma)$ and the requirement of Proposition 4 are satisfied. The minimum step-size is then $\eta_{\min} = \frac{a_{\min}}{2cL_{\max}} = \frac{2a_{\min}(1-\gamma)}{3L_{\max}}$, so $\eta_{\min}$ and $M$ are the same as in the constant step-size case (Theorem 11) and the same rate applies. $\square$

---

**Theorem 13.** *Under the assumptions of Theorem 1 and assuming (iv) non-decreasing preconditioners (v) bounded eigenvalues in the $[a_{\min}, a_{\max}]$ interval, AMSGrad with heavy ball momentum with parameter $\gamma \in [0, 1/4)$, conservative Armijo SLS with $c = 3/4(1-\gamma)$ and uniform averaging*

---

*converges at a rate,*

$$\mathbb{E}[f(\bar{w}_T) - f^*] \le \frac{1}{T}\left(6\frac{1+\gamma^2}{1-4\gamma}L_{\max}D^2\kappa d + \frac{3\gamma}{(1-\gamma)}[f(w_0) - f(w^*)]\right) + 2\sigma^2.$$

*Proof of Theorem 13.* Selecting $c = \sqrt[3]{4(1-\gamma)}$ is feasible if $\gamma < 1/4$ as $c < 1$. The Armijo SLS (Lemma 1) then guarantees that

$$\eta_k \|\nabla f_{i_k}(w_k)\|^2_{A_k^{-1}} \le \frac{1}{c}(f_{i_k}(w_k) - f^*_{i_k}), \qquad \text{and} \qquad \frac{2(1-c)\,a_{\min}}{L_{\max}} \le \eta,$$

which satisfies the requirements of Proposition 4 with $M = \frac{4}{3}(1-\gamma)$. Plugging $M$ in the rate yields

$$\mathbb{E}[f(\bar{w}_T) - f(w^*)] \le \frac{1}{T}\left(6\frac{1+\gamma^2}{1-\gamma}\frac{D^2 a_{\max}d}{\eta_{\min}} + \frac{3\gamma}{(1-\gamma)}[f(w_0) - f(w^*)]\right) + 2\sigma^2,$$

With $c = \frac{3/4}{1-\gamma}$, $\eta_{\min} \ge \frac{2(1-c)a_{\min}}{L_{\max}} = \frac{2a_{\min}}{L_{\max}}\frac{1-4\gamma}{4(1-\gamma)}$. Plugging it into the above bound yields

$$\mathbb{E}[f(\bar{w}_T) - f(w^*)] \le \frac{1}{T}\left(6\frac{1+\gamma^2}{1-4\gamma}L_{\max}D^2\kappa d + \frac{3\gamma}{(1-\gamma)}[f(w_0) - f(w^*)]\right) + 2\sigma^2. \qquad \square$$

We now move to the proof of the main proposition. Our proof follows the structure of Ghadimi et al. (2015); Sebbouh et al. (2020).

*Proof of Proposition 4.* Recall the update for AMSGrad with heavy-ball momentum,

$$w_{k+1} = w_k - \eta_k A_k^{-1}\nabla f_{i_k}(w_k) + \gamma(w_k - w_{k-1}). \tag{12}$$

The proof idea is to analyze the distance from $w^*$ to $w_k$ *and a momentum term*,

$$\|\delta_k\|^2 = \|w_k + m_k - w^*\|^2_{A_k}, \qquad \text{where } m_k = \frac{\gamma}{1-\gamma}(w_k - w_{k-1}), \tag{13}$$

by considering the momentum update (Eq. 12) as a preconditioned step on the joint iterates $(w_k + m_k)$,

$$w_{k+1} + m_{k+1} = w_k + m_k - \frac{\eta_k}{1-\gamma}A_k^{-1}\nabla f_{i_k}(w_k). \tag{14}$$

Let us verify Eq. (14). First, expressing $w_{k+1} + m_{k+1}$ as a weighted difference of $w_{k+1}$ and $w_k$,

$$w_{k+1} + m_{k+1} = w_{k+1} + \frac{\gamma}{1-\gamma}(w_{k+1} - w_k) = \frac{1}{1-\gamma}w_{k+1} - \frac{\gamma}{1-\gamma}w_k.$$

Expanding $w_{k+1}$ in terms of the update rule then gives

$$\begin{aligned} &= \frac{1}{1-\gamma}(w_k - \eta_k A_k^{-1}\nabla f_{i_k}(w_k) + \gamma(w_k - w_{k-1})) - \frac{\gamma}{1-\gamma}w_k, \\ &= \frac{1}{1-\gamma}(w_k - \eta_k A_k^{-1}\nabla f_{i_k}(w_k) - \gamma w_{k-1}), \\ &= \frac{1}{1-\gamma}w_k - \frac{\gamma}{1-\gamma}w_{k-1} - \frac{\eta_k}{1-\gamma}A_k^{-1}\nabla f_{i_k}(w_k), \end{aligned}$$

which can then be re-written as $w_k + m_k - \frac{\eta_k}{1-\gamma}A_k^{-1}\nabla f_{i_k}(w_k)$. The analysis of the method then follows similar steps as the analysis without momentum. Using Eq. (14), we have the recurrence

$$\begin{aligned} \|\delta_{k+1}\|^2_{A_k} = \|w_{k+1} + m_{k+1} - w^*\|^2_{A_k} &= \left\|w_k + m_k - \frac{\eta_k}{1-\gamma}A_k^{-1}\nabla f_{i_k}(w_k) - w^*\right\|^2_{A_k}, \\ &= \|\delta_k\|^2_{A_k} - \frac{2\eta_k}{1-\gamma}\langle \nabla f_{i_k}(w_k), w_k + m_k - w^*\rangle + \frac{\eta_k^2}{(1-\gamma)^2}\|\nabla f_{i_k}(w_k)\|^2_{A_k^{-1}}. \end{aligned} \tag{15}$$

To bound the inner-product, we use Individual Convexity to relate it to the optimality gap,

$$\begin{aligned} \langle \nabla f_{i_k}(w_k), w_k + m_k - w^*\rangle &= \langle \nabla f_{i_k}(w_k), w_k - w^*\rangle + \frac{\gamma}{1-\gamma}\langle \nabla f_{i_k}(w_k), w_k - w_{k-1}\rangle, \\ &\ge f_{i_k}(w_k) - f_{i_k}(w^*) + \frac{\gamma}{1-\gamma}[f_{i_k}(w_k) - f_{i_k}(w_{k-1})], \\ &= \frac{1}{1-\gamma}[f_{i_k}(w_k) - f_{i_k}(w^*)] - \frac{\gamma}{1-\gamma}[f_{i_k}(w_{k-1}) - f_{i_k}(w^*)]. \end{aligned}$$

To bound the gradient norm, we use the step-size assumption that

$$\eta_k \|\nabla f_{i_k}(w_k)\|_{A_k^{-1}}^2 \leq M[f_{i_k}(w_k) - f_{i_k}^*] = M[f_{i_k}(w_k) - f_{i_k}(w^*)] + M[f_{i_k}(w^*) - f_{i_k}^*].$$

For simplicity of notation, let us define the shortcuts

$$h_k(w) = f_{i_k}(w) - f_{i_k}(w^*), \qquad\qquad \sigma_k^2 = f_{i_k}(w^*) - f_{i_k}^*.$$

Plugging those two inequalities in the recursion of Eq. (15) gives

$$\|\delta_{k+1}\|_{A_k}^2 \leq \|\delta_k\|_{A_k}^2 - \frac{\eta_k}{(1-\gamma)^2}(2-M)h_k(w_k) + \frac{2\eta_k\gamma}{(1-\gamma)^2}h_k(w_{k-1}) + \frac{M\eta_k}{(1-\gamma)^2}\sigma_k^2.$$

We can now divide by $\eta_k/(1-\gamma)^2$ and reorganize the inequality as

$$(2-M)h_k(w_k) - 2\gamma h_k(w_{k-1}) \leq \frac{(1-\gamma)^2}{\eta_k}\Big(\|\delta_k\|_{A_k}^2 - \|\delta_{k+1}\|_{A_k}^2\Big) + M\sigma_k^2.$$

Taking the average over all iterations, the inequality yields

$$\frac{1}{T}\sum_{k=1}^{T}(2-M)h_k(w_k) - 2\gamma h_k(w_{k-1}) \leq \frac{1}{T}\sum_{k=1}^{T}\frac{(1-\gamma)^2}{\eta_k}\Big(\|\delta_k\|_{A_k}^2 - \|\delta_{k+1}\|_{A_k}^2\Big) + M\sigma_k^2.$$

To bound the right-hand side, under the assumption that the iterates are bounded by $\|w_k - w^*\| \leq D$, we use Young's inequality to get a bound on $\|\delta_k\|^2$;

$$\|\delta_k\|_2^2 = \|w_k + m_k - w^*\|_2^2 = \left\|\tfrac{1}{1-\gamma}(w_k - w^*) - \tfrac{\gamma}{1-\gamma}(w_{k-1} - w^*)\right\|_2^2$$
$$\leq \frac{2}{(1-\gamma)^2}\Big(\|w_k - w^*\|_2^2 + \gamma^2\|w_{k-1} - w^*\|_2^2\Big) \leq \frac{2(1+\gamma^2)}{(1-\gamma)^2}D^2 = \Delta^2.$$

Given the upper bound $\|\delta_k\|_2 \leq \Delta$, a reorganization of the sum lets us apply Lemma 3 to get

$$\sum_{k=1}^{T}\tfrac{1}{\eta_k}\Big(\|\delta_k\|_{A_k}^2 - \|\delta_{k+1}\|_{A_k}^2\Big) = \sum_{k=1}^{T}\|\delta_k\|_{\frac{1}{\eta_k}A_k}^2 - \sum_{k=1}^{T}\|\delta_{k+1}\|_{\frac{1}{\eta_k}A_k}^2$$
$$= \sum_{k=1}^{T}\|\delta_k\|_{\frac{1}{\eta_k}A_k}^2 - \sum_{k=2}^{T+1}\|\delta_k\|_{\frac{1}{\eta_{k-1}}A_{k-1}}^2$$
$$\leq \sum_{k=1}^{T}\|\delta_k\|_{\frac{1}{\eta_k}A_k}^2 - \sum_{k=1}^{T}\|\delta_k\|_{\frac{1}{\eta_{k-1}}A_{k-1}}^2 + \|\delta_1\|_{\frac{1}{\eta_0}A_0}^2$$
$$= \sum_{k=1}^{T}\|\delta_k\|_{\frac{1}{\eta_k}A_k - \frac{1}{\eta_{k-1}}A_{k-1}}^2 \leq \frac{\Delta^2 a_{\max}d}{\eta_{\min}},$$

where the last step uses the convention $A_0 = 0$ and Lemma 3 on $\delta_k$ instead of $w_k - w^*$. Plugging this inequality in, we get the simpler bound on the right-hand-side

$$\frac{1}{T}\sum_{k=1}^{T}(2-M)h_k(w_k) - 2\gamma h_k(w_{k-1}) \leq \frac{2(1+\gamma^2)D^2 a_{\max}d}{T\eta_{\min}} + M\sigma_k^2.$$

Now that the step-size is bounded deterministically, we can take the expectation on both sides to get

$$\frac{1}{T}\mathbb{E}\left[\sum_{k=1}^{T}(2-M)h(w_k) - 2\gamma h(w_{k-1})\right] \leq \frac{2(1+\gamma^2)D^2 a_{\max}d}{T\eta_{\min}} + M\sigma^2,$$

where $h(w) = f(w) - f^*$ and $\sigma^2 = \mathbb{E}\left[f_{i_k}(w^*) - f_{i_k}^*\right]$. To simplify the left-hand-side, we change the weights on the optimality gaps to get a telescoping sum,

$$\sum_{k=1}^{T}(2-M)h(w_k) - 2\gamma h(w_{k-1}) = \sum_{k=1}^{T}(2-2\gamma-M)h(w_k) + 2\gamma h(w_k) - 2\gamma h(w_{k-1}),$$
$$= (2-2\gamma-M)\left[\sum_{k=1}^{T}h(w_k)\right] + 2\gamma(h(w_T) - h(w_0)),$$
$$\geq (2-2\gamma-M)\left[\sum_{k=1}^{T}h(w_k)\right] - 2\gamma h(w_0).$$

The last inequality uses $h(w_T) \geq 0$. Moving the initial optimality gap to the right-hand-side, we get

$$\frac{1}{T}(2 - 2\gamma - M)\,\mathbb{E}\left[\sum_{k=1}^{T} h(w_k)\right] \leq \frac{1}{T}\left(\frac{2(1+\gamma^2)D^2 a_{\max}d}{\eta_{\min}} + 2\gamma h(w_0)\right) + M\sigma^2.$$

Assuming $2 - 2\gamma - M > 0$ and dividing, we get

$$\frac{1}{T}\,\mathbb{E}\left[\sum_{k=1}^{T} h(w_k)\right] \leq \frac{1}{2 - 2\gamma - M}\left[\frac{1}{T}\left(\frac{2(1+\gamma^2)D^2 a_{\max}d}{\eta_{\min}} + 2\gamma h(w_0)\right) + M\sigma^2\right].$$

Using Jensen's inequality and averaging the iterates finishes the proof. $\qquad\square$

## F  EXPERIMENTAL DETAILS

Our proposed adaptive gradient methods with SLS and SPS step-sizes are presented in Algorithms 1 and 3. We now make a few additional remarks on the practical use of these methods.

---

**Algorithm 1** Adaptive methods with SLS($f$, `precond`, $\beta$, `conservative`, `mode`, $w_0$, $\eta_{\max}$, $b$, $c \in (0,1)$, $\gamma < 1$)

---

1: **for** $k = 0, \ldots, T-1$ **do**
2:      $i_k \leftarrow$ sample mini-batch of size $b$
3:      $A_k \leftarrow$ `precond`$(k)$                                               ▷ Form the preconditioner.
4:      **if** `mode` == `Lipschitz` **then**
5:          $p_k \leftarrow \nabla f_{i_k}(w_k)$
6:      **else if** `mode` == `Armijo` **then**
7:          $p_k \leftarrow A_k^{-1} \nabla f_{i_k}(w_k)$
8:      **end if**
9:      **if** `conservative` **then**
10:          **if** k == 0 **then**
11:              $\eta_k \leftarrow \eta_{\max}$
12:          **else**
13:              $\eta_k \leftarrow \eta_{k-1}$
14:          **end if**
15:      **else**
16:          $\eta_k \leftarrow \eta_{\max}$
17:      **end if**
18:      **while** $f_{i_k}(w_k - \eta_k \cdot p_k) > f_{i_k}(w_k) - c\,\eta_k \langle \nabla f_{i_k}(w_k), p_k \rangle$ **do**      ▷ Line-search loop
19:          $\eta_k \leftarrow \gamma\,\eta_k$
20:      **end while**
21:      $m_k \leftarrow \beta m_{k-1} + (1-\beta)\nabla f_{i_k}(w_k)$
22:      $w_{k+1} \leftarrow w_k - \eta_k A_k^{-1} m_k$
23: **end for**
24: **return** $w_T$

---

---

**Algorithm 2** reset($\eta$, $\eta_{\max}$, $k$, $b$, $n$, $\gamma$, `opt`)

---

1: **if** $k = 0$ **then**
2:      **return** $\eta_{\max}$
3: **else if** `opt` $= 0$ **then**
4:      $\eta \leftarrow \eta$
5: **else if** `opt` $= 1$ **then**
6:      $\eta \leftarrow \eta \cdot \gamma^{b/n}$
7: **else if** `opt` $= 2$ **then**
8:      $\eta \leftarrow \eta_{\max}$
9: **end if**
10: **return** $\eta$

---

As suggested by Vaswani et al. (2019b), the standard backtracking search can sometimes result in step-sizes that are too small while taking bigger steps can yield faster convergence. To this end, we adopted their strategies to reset the initial step-size at every iteration (Algorithm 2). In particular, using reset option 0 corresponds to starting every backtracking line search from the step-size used in the previous iteration. Since the backtracking never increases the step-size, this option enables the "conservative step-size" constraint for the Lipschitz line-search to be automatically satisfied. For the Armijo line-search, we use the heuristic from Vaswani et al. (2019b) corresponding to reset option 1. This option begins every backtracking with a slightly larger (by a factor of $\gamma^{b/n}$, $\gamma = 2$ throughout our experiments) step-size compared to the step-size at the previous iteration, and works well consistently across our experiments. Although we do not have theoretical guarantees for Armijo

SLS with general preconditioners such as Adam, our experimental results indicate that this is in fact a promising combination that also performs well in practice.

---

**Algorithm 3** Adaptive methods with SPS($f$, $[f_i^*]_{i=1}^n$, `precond`, $\beta$, `conservative`, `mode`, $w_0$, $\eta_{\max}$, $b$, $c$)

---

1: **for** $k = 0, \ldots, T-1$ **do**
2:      $i_k \leftarrow$ sample mini-batch of size $b$
3:      $A_k \leftarrow \texttt{precond}(k)$                                         ▷ Form the preconditioner
4:      **if** `mode == Lipschitz` **then**
5:          $p_k \leftarrow \nabla f_{i_k}(w_k)$
6:      **else if** `mode == Armijo` **then**
7:          $p_k \leftarrow A_k^{-1} \nabla f_{i_k}(w_k)$
8:      **end if**
9:      **if** `conservative` **then**
10:          **if** k == 0 **then**
11:              $\eta_B \leftarrow \eta_{\max}$
12:          **else**
13:              $\eta_B \leftarrow \eta_{k-1}$
14:          **end if**
15:      **else**
16:          $\eta_B \leftarrow \eta_{\max}$
17:      **end if**
18:      $\eta_k \leftarrow \min \left\{ \frac{f_{i_k}(w_k) - f_{i_k}^*}{c \langle \nabla f_{i_k}(w_k), p_k \rangle}, \eta_B \right\}$
19:      $m_k \leftarrow \beta m_{k-1} + (1-\beta) \nabla f_{i_k}(w_k)$
20:      $w_{k+1} \leftarrow w_k - \eta_k A_k^{-1} m_k$
21: **end for**
22: **return** $w_T$

---

On the other hand, rather than being too conservative, the step-sizes produced by SPS between successive iterations can vary wildly such that convergence becomes unstable. Loizou et al. (2020) suggested to use a smoothing procedure that limits the growth of the SPS from the previous iteration to the current. We use this strategy in our experiments with $\tau = 2^{b/n}$ and show that both SPS and Armijo SPS work well. For the convex experiments, for both SLS and SPS, we set $c = 0.5$ as is suggested by the theory. For the non-convex experiments, we observe that all values of $c \in [0.1, 0.5]$ result in reasonably good performance, but use the values suggested in Vaswani et al. (2019b); Loizou et al. (2020), i.e. $c = 0.1$ for all adaptive methods using SLS and $c = 0.2$ for methods using SPS.

# G    ADDITIONAL EXPERIMENTAL RESULTS

In this section, we present additional experimental results showing the effect of the step-size for adaptive gradient methods using a synthetic dataset (Fig. 4). We show the wall-clock times for the optimization methods (Fig. 5). We show the variation in the step-size for the SLS methods when training deep networks for both the CIFAR in Fig. 6 and ImageNet (Fig. 7) datasets. We evaluate these methods on easy non-convex objectives - classification on MNIST (Fig. 8) and deep matrix factorization (Fig. 10). We use deep matrix factorization to examine the effect of over-parameterization on the performance of the optimization methods and check the methods' performance when minimizing convex objectives associated with binary classification using RBF kernels in Fig. 9. Finally in Fig. 11, we quantify the gains of incorporating momentum in AMSGrad by comparing against the performance AMSGrad *without momentum*.

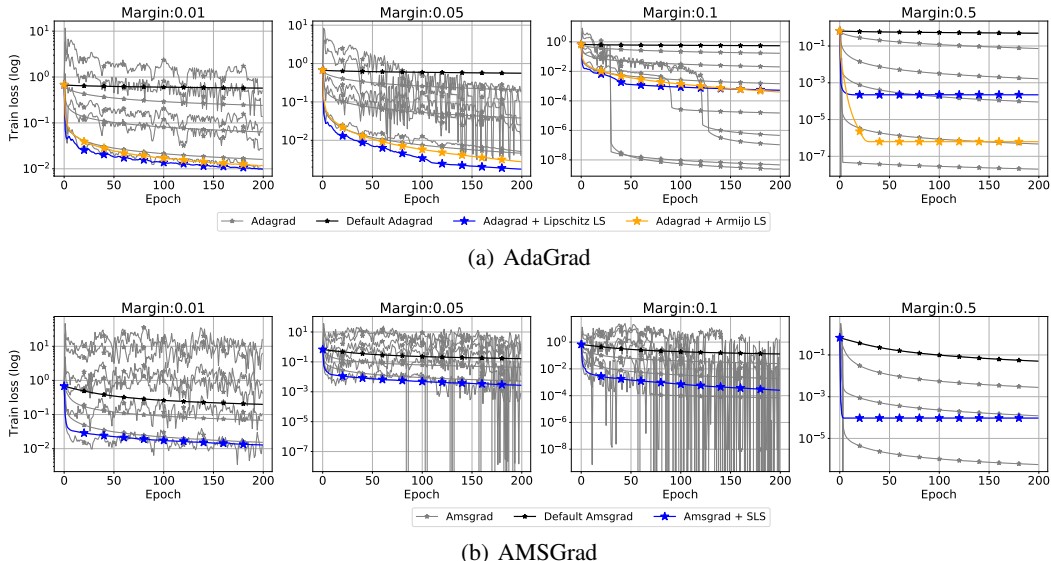

Figure 4: Effect of step-size on the performance of adaptive gradient methods for binary classification on a linearly separable synthetic dataset with different margins. We observe that the large variance for the adaptive gradient methods, and the variants with SLS have consistently good performance across margins and optimizers.

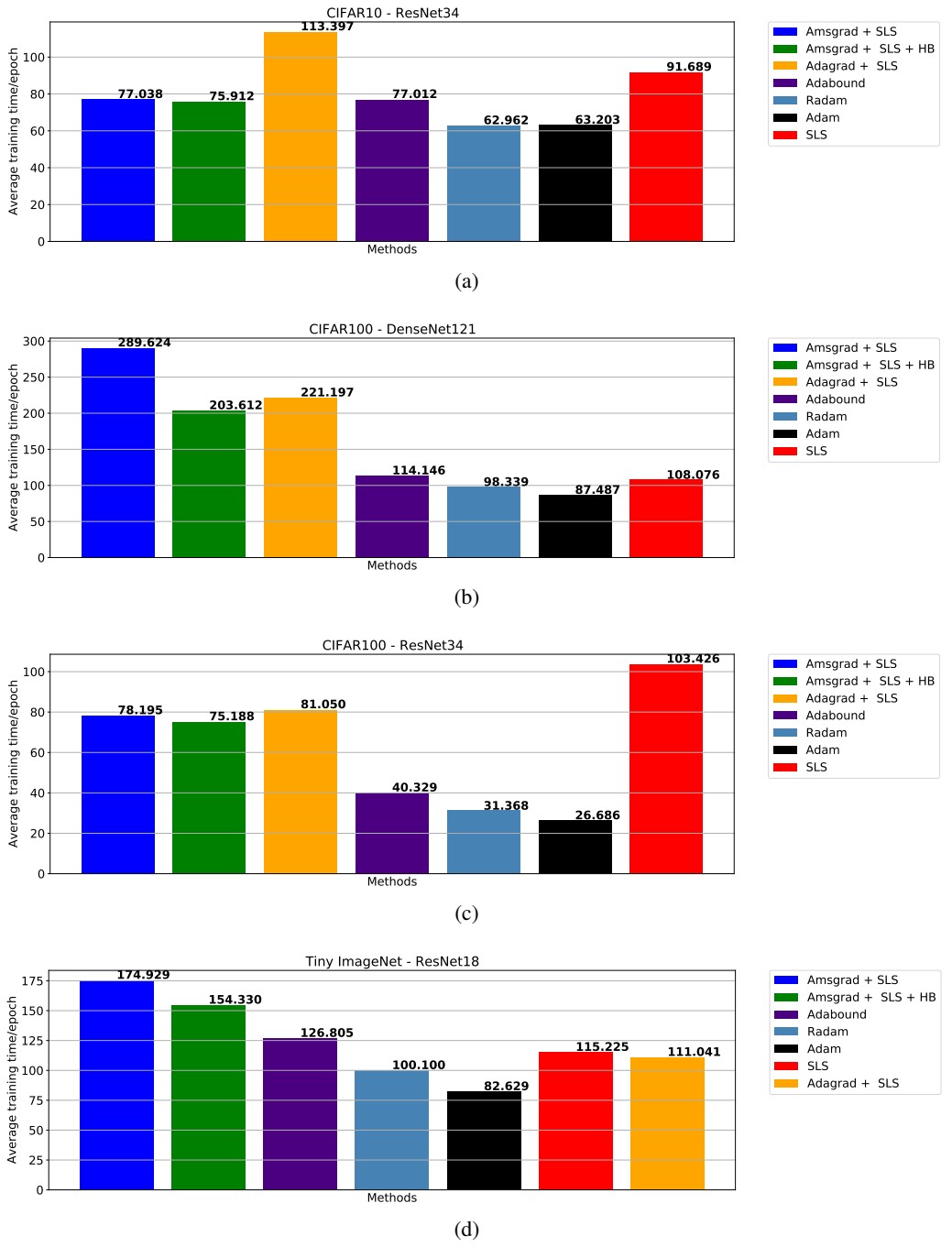

Figure 5: Runtime (in seconds/epoch) for optimization methods for multi-class classification using the deep network models in Fig. 2. Although the runtime/epoch is larger for the SLS/SPS variants, they require fewer epochs to reach the maximum test accuracy (Figure 2). This justifies the moderate increase in wall-clock time.

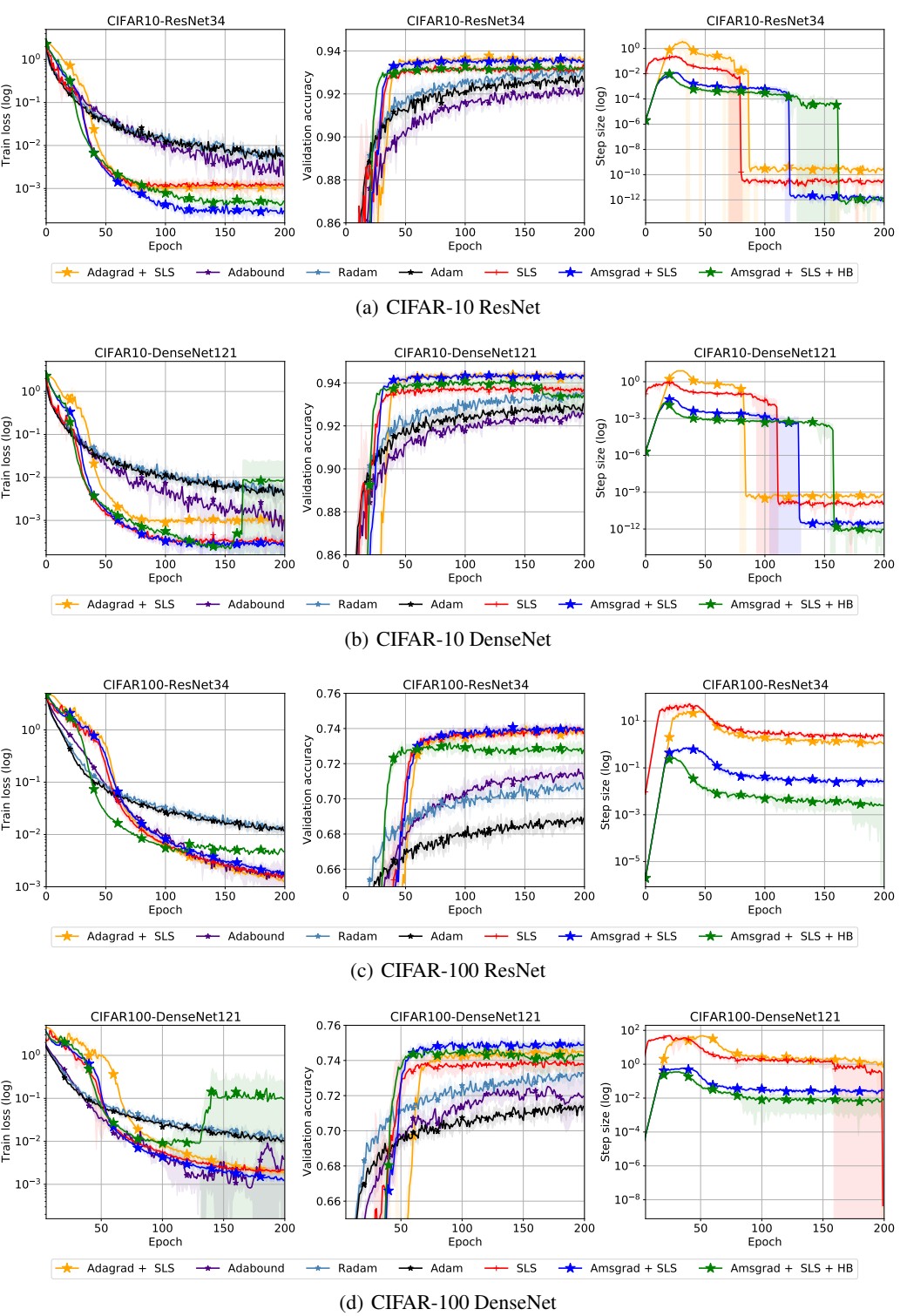

Figure 6: Comparing optimization methods on image classification tasks using ResNet and DenseNet models on the CIFAR-10/100 datasets. For the SLS/SPS variants, refer to the experimental details in Appendix F. For Adam, we did a grid-search and use the best step-size. We use the default hyper-parameters for the other baselines. We observe the consistently good performance of AdaGrad and AMSGrad with Armijo SLS. We also show the variation in the step-size and observe a cyclic pattern (Loshchilov & Hutter, 2017) - an initial warmup in the learning rate followed by a decrease or saturation to a small step-size (Goyal et al., 2017).

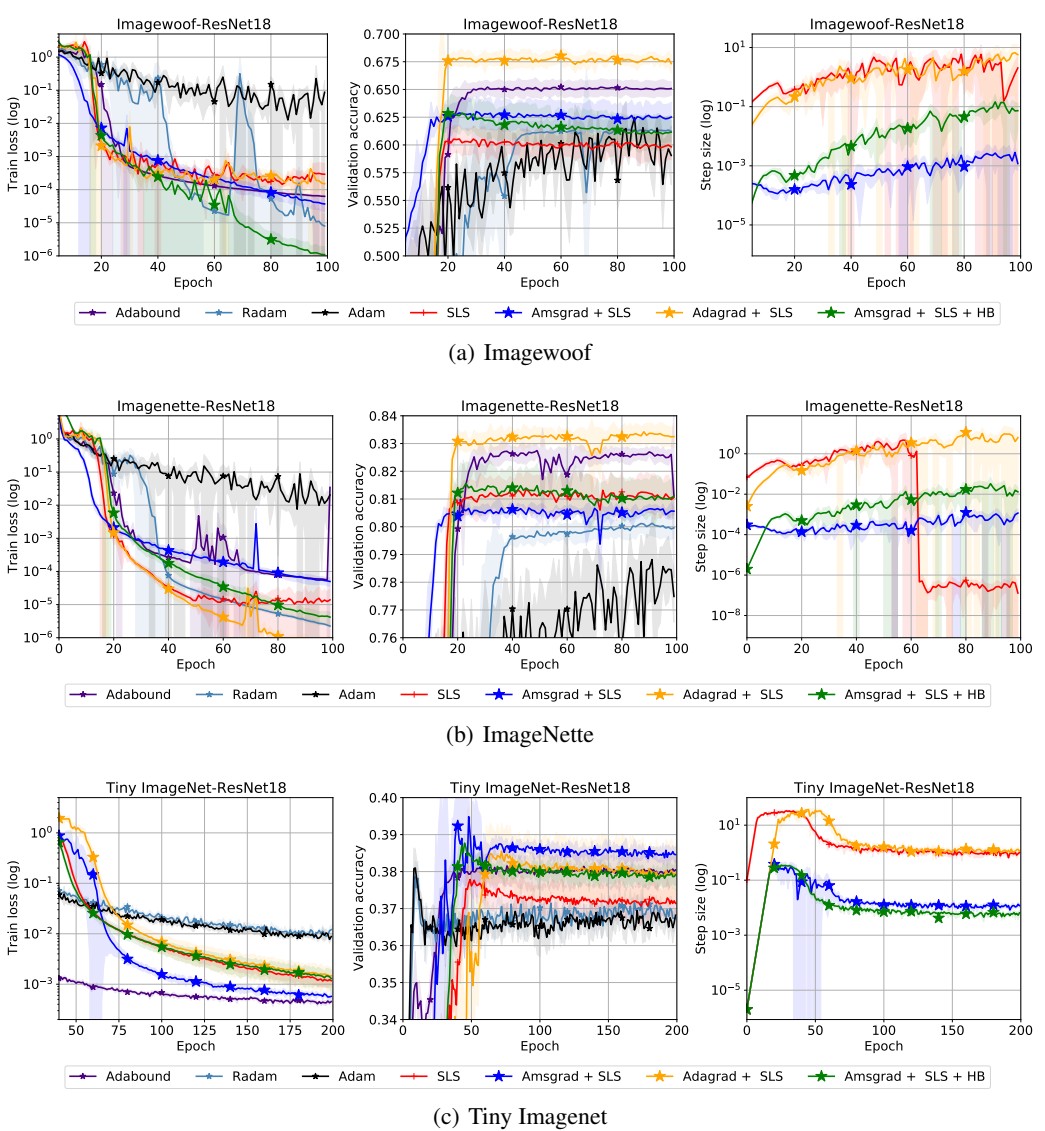

Figure 7: Comparing optimization methods on image classification tasks using variants of ImageNet. We use the same settings as the CIFAR datasets and observe that AdaGrad and AMSGrad with Armijo SLS is consistently better.

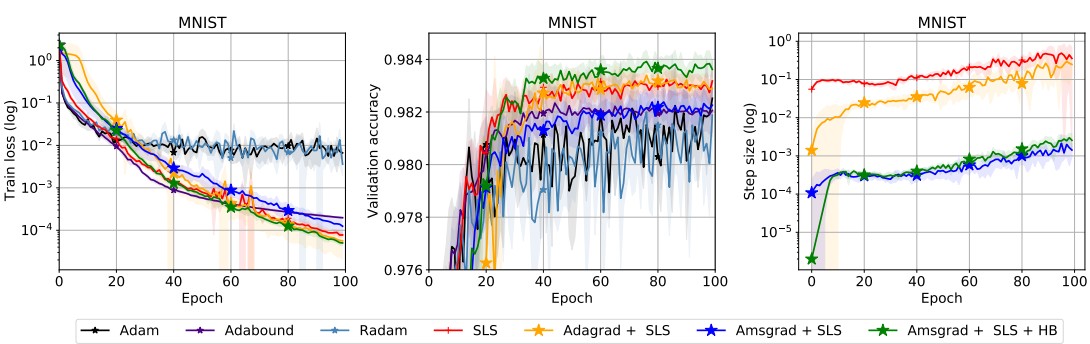

Figure 8: Comparing optimization methods on MNIST.

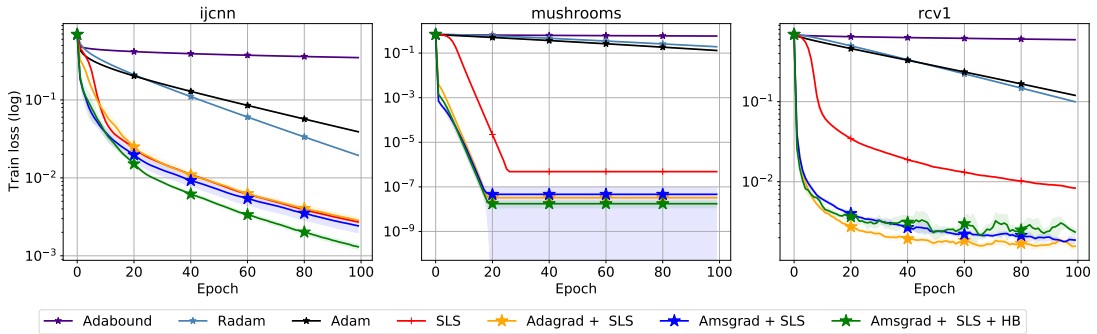

Figure 9: Comparison of optimization methods on convex objectives: binary classification on LIBSVM datasets using RBF kernel mappings. The kernel bandwidths are chosen by cross-validation following the protocol in (Vaswani et al., 2019b). All line-search methods use $c = 1/2$ and the procedure described in Appendix F. The other methods are use their default parameters. We observe the superior convergence of the SLS variants and the poor performance of the baselines.

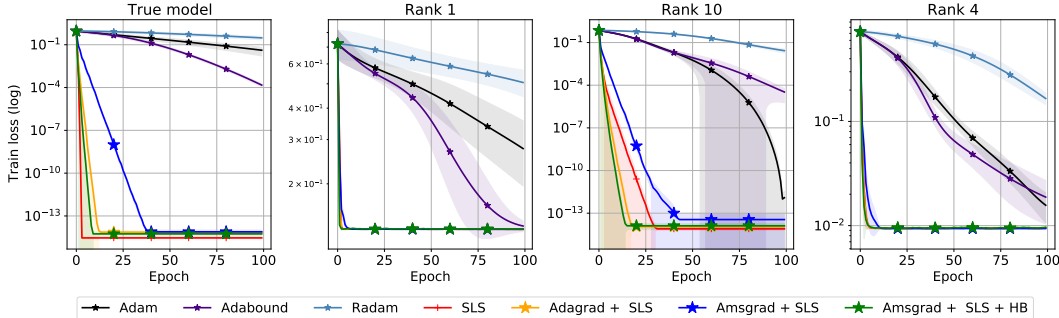

Figure 10: Comparison of optimization methods for deep matrix factorization. Methods use the same hyper-parameter settings as above and we examine the effects of over-parameterization on the problem: $\min_{W_1, W_2} \mathbb{E}_{x \sim N(0,I)} \|W_2 W_1 x - Ax\|^2$ (Vaswani et al., 2019b; Rolinek & Martius, 2018). We choose $A \in \mathbb{R}^{10 \times 6}$ with condition number $\kappa(A) = 10^{10}$ and control the over-parameterization via the rank $k$ (equal to 1, 4, 10) of $W_1 \in \mathbb{R}^{k \times 6}$ and $W_2 \in \mathbb{R}^{10 \times k}$. We also compare against the true model. In each case, we use a fixed dataset of 1000 samples. We observe that as the over-parameterization increases, the performance of all methods improves, with the methods equipped with SLS performing the best.

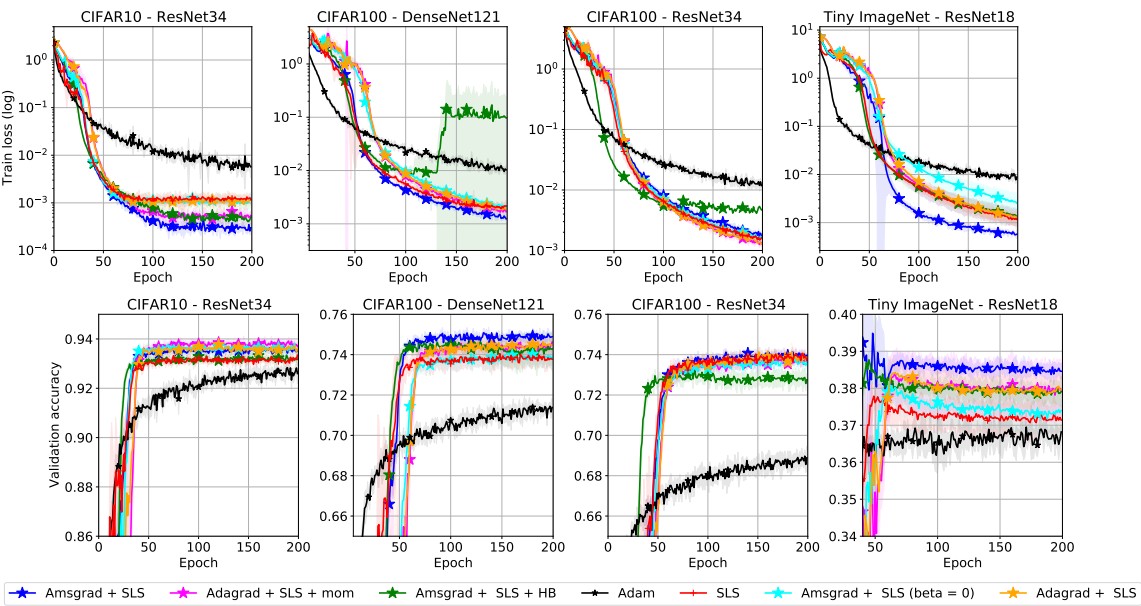

Figure 11: Ablation study comparing variants of the basic optimizers for multi-class classification with deep networks. Training loss (top) and validation accuracy (bottom) for CIFAR-10, CIFAR-100 and Tiny ImageNet. We consider the AdaGrad with AMSGrad-like momentum and do not find improvements in performance. We also benchmark the performance of AMSGrad without momentum, and observe that incorporating AMSGrad momentum does improve the performance, whereas heavy-ball momentum has a minor, sometimes detrimental effect. We use SLS and Adam as benchmarks to study the effects of incorporating preconditioning vs step-size adaptation.

