# OpenReview forum: "Adaptive Gradient Methods Converge Faster with Over-Parameterization (and you can do a line-search)"
_ICLR.cc/2021/Conference — Reject_

### Official Review · AnonReviewer1 · 2020-10-27
**Recommendation to Accept**

**Rating:** 5
**Confidence:** 4

**Review:**

This paper revisited two important stochastic algorithms, AdaGrad and AMSGrad. It reanalyzes these two algorithms under interpolation setup and shows how the results get improved under this particular case.
***
Strength: interpolation setup is reasonable in over-parametrized regime. This paper establishes a thorough analysis on AdaGrad and AMSGrad under the interpolation setup. Further, it incorporates the stochastic line search technique and stochastic Polyak stepsize technique with AMSGrad to make the stepsize selection adaptive. The improvement results and the dependence on the extent of interpolation violation seem interesting.
***
Concerns: why do the authors not consider the unconservative stochastic line search and Polyak for AdaGrad. Also, I'm wondering what's the technical challenges to reestablish these results. Is that simply combining the classical analysis with the previous work (Vaswani et al. 2019 Painless stochastic gradient: Interpolation, line-search, and convergence rates). The stepsize lies in a bounded interval lower bounded away from zero, so that the randomness of stepsize is well controlled.
***
For future improvement, I think authors should emphasize the difficulty of the analysis more clear. Based on my reading, authors list comprehensive results while how significance of these results is less discussed. It would be good to provide a general proof framework to make "how interpolation helps on a sharper analysis" more clear.


***
During rebuttal:
The authors should highlight some technical difficulties in the paper. In principle, stochastic line search under overparameterized regime does not make things harder because the stepsize is lower bounded. The difficulty of stochastic stepsize is to control the product of stepsize and gradient, while under this regime the product is separable. It seems the analysis of momentum plus this observation is enough for the analysis. It would be useful to provide some insights and challenges of the analysis.

---

> ### Author Response · Authors · 2020-11-17
> **Clarifications during rebuttal**
>
> We thank the reviewer for their useful feedback and suggestions. Here, we address the concerns listed above. The necessary changes have been made (in red) in the rebuttal revision.
>
> 1. **Why do the authors not consider the unconservative stochastic line search and Polyak for AdaGrad.**
> The unconservative stochastic line-search for AdaGrad is considered in Appendix C.3. Similar to AMSGrad, this would result in an $O(1/T)$ convergence to the solution when interpolation is exactly satisfied. However, when interpolation is not satisfied exactly, AdaGradwith an unconservative step-size will converge to a neighborhood of the solution, rather than the exact solution as is the case for the conservative step-size.
>
> 2. **I'm wondering what's the technical challenges to reestablish these results. Is that simply combining the classical analysis with the previous work (Vaswani et al. 2019 Painless stochastic gradient: Interpolation, line-search, and convergence rates). The stepsize lies in a bounded interval lower bounded away from zero, so that the randomness of stepsize is well controlled.**
> One of the technical challenges over (Vaswani et al. 2019) is to use the line-search/Polyak step-size techniques with momentum. Using momentum significantly complicates the analyses. Please refer to Appendix E for the proofs corresponding to the two types of momentum we considered. In fact, our theoretical results justify the use of the heuristic used to incorporate momentum in (Vaswani et al. 2019).
> Furthermore, in order to derive their convergence results, Vaswani et al assumed that interpolation is exactly satisfied. Another technical challenge was to show a ``smooth'' degradation of the convergence rates when interpolation is not exactly satisfied. Cevher et al 2019 address this for a constant step-size in "On the linear convergence of the stochastic gradient method with constant step-size" and we show that such a smooth degradation is also possible with a line-search.

---

### Official Review · AnonReviewer4 · 2020-10-28
**concerns about theory.**

**Rating:** 5
**Confidence:** 4

**Review:**

This paper analyzes adaptive algorithms such as adagrad and AMSGrad in a finite-sum optimization problem. The proofs appear to treat this setting through online convex optimization and online-to-batch conversion. It is shown that both AdaGrad and AMSGrad improve when the individual losses are all minimized at the same point. Further, line search techniques are analyzed in conjunction with these algorithms, and empirical results show that in practice the line searches speed up convergence.

I do not think Theorem 1 is particularly novel. I am not sure of an original reference (it may be a kind of folklore), but see for example https://parameterfree.com/2019/09/20/adaptive-algorithms-l-bounds-and-adagrad/ theorem 7, from which it is trivial to deduce Theorem 1 by observing that WLOG we may assume f_i^*=0 for all i since subtracting the minimum value does not change the gradients.

For Theorem 2, I may be missing something. This value of alpha seems *strictly worse* than what we would get in Theorem 1 by just setting eta=eta_max. So what is the line search buying us? Is it just for empirical performance with no theoretical benefit yet? It is not obviously presented this way, so if so I think some remarks to this effect are in order.

The assumption on bounded eigenvalues for the results on AMSGrad seems a little troublesome to me: I am worried that all of the adaptive nature of the preconditioner is irrelevant and these assumptions are doing all the heavy-lifting. Indeed, if we set \beta=0, then with learning rate \eta = a_min/L and sgd update w_{t+1} = w_t - eta * V^(-1) g_t for *any* V in [a_min, a_max], then I suspect that standard analysis of gradient descent using learning rates at most 1/L will yield fairly similar results to Theorem 3. I would be happy to hear otherwise, though!

My overall feeling is that there is a missing piece here in the theory to show that the line search is useful. I am not confident that the other results are significant on their own.

As for the empirical results, these seem like reasonable gains over Adam. I would have preferred to see more standard deep learning benchmarks on non-image tasks as well,, but I am not an expert here and so would defer to other opinions.


Nits:
I am not sure that the assumption that the iterates are bounded is well justified here. I do believe it has been assumed in some past literature, but this does not make it actually true. It is certainly *not* standard in the literature on online learning. The two references (Duchi et al 2011 and Levy et al 2018) cited as evidence here do not actually assume this. Instead, they use projections to *ensure* that the iterates are bounded without assumptions.

---

> ### Author Response · Authors · 2020-11-16
> **Clarifications during rebuttal - Part 1**
>
> We thank the reviewer for their useful feedback and suggestions. Here, we address the concerns listed above. The necessary changes have been made (in red) in the rebuttal revision.
>
> 1. **I do not think Theorem 1 is particularly novel. I am not sure of an original reference (it may be a kind of folklore), but see for example https://parameterfree.com/2019/09/20/adaptive-algorithms-l-bounds-and-adagrad/ theorem 7, from which it is trivial to deduce Theorem 1 by observing that WLOG we may assume $f_i^\star=0$ for all $i$ since subtracting the minimum value does not change the gradients.**
> To the best of our knowledge, this result is indeed folklore and does not appear in the literature. We were not aware of this blog-post, and agree that Theorem 1 is similar to the result you pointed out. We have now mentioned it in Section 3 of the paper. Our theorem slightly generalizes this result to use a matrix preconditioner (instead of the scalar version of AdaGrad) and works for any step-size (instead of the specific step-size used in Theorem 7 of this result). More importantly, it identifies a practical application (over-parameterized models) where this result is useful.
>
> 2. **For Theorem 2, I may be missing something. This value of alpha seems strictly worse than what we would get in Theorem 1 by just setting $eta=eta_{max}$. So what is the line search buying us? Is it just for empirical performance with no theoretical benefit yet? It is not obviously presented this way, so if so I think some remarks to this effect are in order.**
> Yes, the claim is that the line-search improves the empirical performance while retaining the favourable *worst-case* convergence guarantees of AdaGrad. We have clarified this in the paper. Please refer to Figure 1 to observe the empirical effect of using the line-search vs choosing an arbitrary value of $\eta$.
>
> 3. **The assumption on bounded eigenvalues for the results on AMSGrad seems a little troublesome to me: I am worried that all of the adaptive nature of the preconditioner is irrelevant and these assumptions are doing all the heavy-lifting. Indeed, if we set $\beta=0$, then with learning rate $\eta = a_{min}/L$ and sgd update $w_{t+1} = w_t - \eta  V^{-1} g_t$ for any $V$ in $[a_{min}, a_{max}]$, then I suspect that standard analysis of gradient descent using learning rates at most $1/L$ will yield fairly similar results to Theorem 3. I would be happy to hear otherwise, though!**
> Unlike AdaGrad, the AMSGrad preconditioner does not have nice properties enabling a better analysis. Consequently, the existing analyses for AMSGrad assume that the preconditioner eigenvalues are bounded. Since we focus on the effect of the step-size, momentum, and over-parameterization for the existing methods, we made this simplifying assumption as well.
> From a technical perspective, in the absence of momentum, we agree that assuming bounded preconditioners makes the analysis easier and close to SGD. However, the analysis becomes much more complicated when incorporating momentum. In fact, ours is the first analysis of AMSGrad with a constant momentum and step-size (which is what is done in practice). The original analysis Reddi et al, 2019 used both a decreasing step-size and momentum and recently, Alacaoglu et al, 2020 uses a constant momentum, but a decreasing step-size in order to easily bound the resulting momentum terms. Furthermore, using line-search techniques to automatically set the step-size for AMSGrad with momentum is novel and non-trivial. As a matter of fact, such a result was not known even for SGD with momentum.
>
> 4. **My overall feeling is that there is a missing piece here in the theory to show that the line search is useful. I am not confident that the other results are significant on their own.**
> While we would also like to show that line-search improves the worst-case performance, such a result does not exist even for deterministic gradient descent. Even in that case, Armijo line-search matches the rates of GD with the best constant step-size (that depends on the smoothness) but can vastly out-perform it in practice.
> Similar to the results for deterministic GD, our theoretical analysis shows that it is possible for line-search techniques to match the convergence rates of AMSGrad with the best constant step-size. Specifically, the line-search matches these rates *without* the knowledge of $a_{\min}$ or $L_{\max}$. For AdaGrad, line-search retains the favorable convergence properties while significantly improving the practical performance.

---

> ### Author Response · Authors · 2020-11-16
> **Clarifications during rebuttal - Part 2**
>
> 5. **As for the empirical results, these seem like reasonable gains over Adam. I would have preferred to see more standard deep learning benchmarks on non-image tasks as well,, but I am not an expert here and so would defer to other opinions.**
> The empirical gains over Adam are substantial and consistent over the whole range of convex and non-convex benchmarks we consider. Please refer to Appendix G for additional experimental results. We believe that consistently out-performing the most used optimization method across a range of tasks is a significant contribution.
>
> 6. **I am not sure that the assumption that the iterates are bounded is well justified here. I do believe it has been assumed in some past literature, but this does not make it actually true. It is certainly not standard in the literature on online learning. The two references (Duchi et al 2011 and Levy et al 2018) cited as evidence here do not actually assume this. Instead, they use projections to ensure that the iterates are bounded without assumptions.**
> It is possible to modify our proofs to include a projection step that ensures the iterates are bounded. For example, this would require using Lemma 4 in Appendix G from Reddi et al (2019) at the beginning of the proof, while the rest of the proof would mostly remain unchanged. We have clarified this in Section 2 of the paper.
> We made the bounded iterates assumption to avoid this additional complication in what are already complicated proofs. We emphasize that the goal of our theoretical results is to understand the effect of the step-size, momentum and over-parameterization on the performance of adaptive gradient methods. It is not to resolve all theoretical issues in the analysis of adaptive gradient methods. Although popular, these methods are poorly understood and we need some simplifying assumptions to focus on the things that influence their empirical performance.

---

### Official Review · AnonReviewer2 · 2020-10-29
**Reviewer Summary**

**Rating:** 6
**Confidence:** 4

**Review:**

This paper studies adaptive gradient methods under the over-parametrized settings, where the authors study the converge in the interpolation setting. In this setting, the optimal objective is 0. The authors show that the convergence rate is O(1/T). In addition, when the interpolation is approximately satisfied, the authors show the convergence to a neighborhood of the solution. The authors also provide theoretical justifications for popular line search methods.

Overall, I find the paper easy to read. However, I do have a few questions that would like to see the authors' answers:

1. The authors implicitly assume that the optimal solution is unique.  However, this is not the case in many over-parametrized models. For example, consider the case for logistic regression where the two classes are perfectly separable. The minimizer is not well defined, but there have been extensive work on this topic. Can the authors' analysis adapt to such situations?

2. Taking the logistic regression as an example again, the "minimizer" is not within a bounded region. Can the authors' analysis been adopted to analyze such case?

3. The result are all in the form of expectation. Can the authors bound the L2-norm?

4. I think it would be informative to add the result when the loss is strongly convex, where we can have the bound for the solutions.

---

> ### Author Response · Authors · 2020-11-16
> **Clarifications during rebuttal**
>
> We thank the reviewer for their useful feedback and suggestions. Here, we address the questions listed above. The necessary changes have been made (in red) in the rebuttal revision.
>
> 1. While the text refers to an optimal solution $w^*$ for simplicity of presentation, it need not be unique. Any $w^* \in \arg\min_w f(w)$ would satisfy the necessary properties. Our theoretical results bound the suboptimality in the function $f(\bar{w}_T) - f(w^*)$, which is uniquely defined even if they have multiple minima. We have now clarified this in Section 2.
>
> 2. Unlike other losses that have finite solutions, for the logistic loss, the minimizer is at infinity. In this case, we can modify our analyses to have a projection step onto a ball of radius $D$, ensuring that the iterates do remain bounded. However, perfect interpolation would not possible in this case and the methods will converge to a neighbourhood of the solution that depends on $D$. Note that the standard SGD analysis that depends on $\vert \vert w_0 - w^* \vert \vert$ will have a similar issue.
>
> 3. We understand that the question refers to bounds in high-probability rather than in expectation. Please correct us if our understanding of your question is incorrect. We believe that we can adopt the analysis in "A new regret analysis
> for adam-type algorithms", Alacaoglu et al., 2020 to prove high probability bounds on the suboptimality, but do not think such an analysis will yield new insights into the problem. We thus leave this for future work.
>
> 4. Indeed, for strongly convex functions, distance to the optimum $\vert \vert w_k - w^* \vert \vert^{2}$  is upper-bounded by the difference in the function values, and we do not need the bounded iterates assumption. However, note that this only results in an $O(1/T + \sigma^2)$ rate instead of the linear rate $\exp(-T) + \sigma^2$ expected for strongly-convex functions. In general, proving linear convergence rates for adaptive methods is difficult (there is only a very recent result for AdaGrad in "Linear Convergence of Adaptive Stochastic Gradient Descent", Xie et al, 2020).

---

### Official Review · AnonReviewer3 · 2020-11-03
**Nice insight with difficult-to-check assumptions**

**Rating:** 7
**Confidence:** 4

**Review:**

##########################################################################


Summary:

This paper studies the convergence of adaptive gradient methods under an interpolation assumption, showing for example these methods can converge at an O(1/t), instead of O(1/\sqrt{t}) rate when perfect interpolation is satisfied. Convergence behaviors with line search and Polyak step sizes are also analyzed.


##########################################################################


Reasons for score:


This paper provides good insights regarding how an interpolation assumption may help accelerate adaptive gradient methods. I do not feel the technical results very solid, as some difficult-to-check properties are just put as assumptions (see cons).


##########################################################################Pros:


Pros:

1. The results provides insights regarding why adaptive gradient methods may converge faster when the interpolation assumption is satisfied.

2. Line search and Polyak step size methods help address the need of problem and algorithm parameters in standard theories. Moreover, there are few papers discussing line search and Polyak step size methods in the finite-sum setup.

3. The Polyak step size is well-motivated in the interpolation setting.


##########################################################################

Cons:

1. The abstract claims that “AdaGrad can achieve an O(1) regret in the online convex optimization framework.” I do not see this result in the main text.

2. The paper reads waiving regarding difficult-to-check assumptions. In particular, this paper assumes the sequence of iterates is bounded in a set of radius D and the eigenvalue of the preconditioning matrices are bounded. The main argument supporting these assumptions are simply they are “common” in existing literature. I feel the technical challenges in the analyses are alleviated a lot because of these “common” assumptions.   Notice that without the conditions, the convergence guarantees may not be meaningful because the D parameter in all theorems and a_{min} and a_{max} in Theorem 3 and Theorem 4 can scale with the iteration counter.

3. The proposed line search methods seem to be computationally very expensive. The proposed line search methods require computing the largest step size satisfying the desired inequality, instead of the *largest among a sequence of exponentially decaying step sizes* as in standard Armijo line search methods. Is it possible to analyze the performance of latter—more computationally favorable—scheme?

4. It is claimed that the step size chosen by the proposed conservative Lipschitz line search method is bounded in [2 (1 - c) / L_{max}, \eta_{k - 1}]. Can it happen that \eta_{k - 1} \leq 2 (1 - c) / L_{max}? If yes, then is the step size selection rule well-defined? There is also a similar claim for the stochastic Armijo line search method.

5. I don’t get the sentence that “a similar distinction between the convergence of constant step-size Adam (or AMSGrad) vs. AdaGrad has also been recently discussed in the non-convex setting (Défossez et al., 2020)” in Section 4. What is the distinction?

6. Minor comment:
    1. Typo in the first paragraph: online batch reduction -> online-to-batch reduction
    2. m_1 is not specified in (1).
    3. The abbreviation SPS is not defined when it appears for the first time in the main text.

##########################################################################

After reading author rebuttal:

I think the value of this work is to demonstrate the possible benefits of an interpolation assumption. Hence, though there are some theoretical issue and theory-practice gap, I keep the original score.

- I do not feel it very reasonable to emphasize the regret result in the abstract and then put the corresponding section in the appendix. It is more reasonable to move Appendix C.2 to the main text though I guess it would be difficult in practice.

- I understand adding a projection step typically does not change the proof much. However, this is not the setup analyzed in this paper. I feel the associated newly-added clarification in Section 2 reads somewhat waiving. I suggest the authors add some clarifications in the revision similar to the following in the rebuttal.

"Without an explicit projection step, we believe it is possible to adopt the recent SGD analysis "On the almost sure convergence of stochastic gradient descent in non-convex problems, NeurIPS, 2020." to prove that the iterates do remain bounded with high-probability. We leave this for future work."

- The other clarifications are OK to me.

- I suggest the authors add some clarification regarding 4 in the main text.


#########################################################################

---

> ### Author Response · Authors · 2020-11-16
> **Clarifications during rebuttal**
>
> We thank the reviewer for their useful feedback and suggestions. Here, we address and clarify the cons listed above. The necessary changes have been made (in red) in the rebuttal revision.
>
> 1. We did not include the regret bound in the main text as its proof is similar to Theorem 1. The result and its corresponding proof is in Appendix C.2. This is mentioned in the introduction, but we have now clarified it after Theorem 1.
>
> 2. Note that the bounded eigenvalues condition can be enforced algorithmically.  As explained in Section 3, ``for diagonal preconditioners, such a boundedness property is easy to verify, and it is also inexpensive to maintain the desired range by projection''. For the assumption on the bounded iterates, it is possible to modify our proofs to include a projection step that ensures this property. For example, this would require using Lemma 4 in Appendix G from Reddi et al (2019) at the beginning of the proof, while the rest of the proof would mostly remain unchanged. We have added a clarification to Section 2 of the paper. Without an explicit projection step, we believe it is possible to adopt the recent SGD analysis "On the almost sure convergence of stochastic gradient descent in non-convex problems, NeurIPS, 2020." to prove that the iterates do remain bounded with high-probability. We leave this for future work.
> We emphasize that the goal of our theoretical results is to understand the effect of the step-size, momentum, and over-parameterization on the performance of adaptive gradient methods. It is not to resolve all theoretical issues in the analysis of adaptive gradient methods. Although popular, these methods are poorly understood and we need some simplifying assumptions to focus on the things that influence their empirical performance.
>
> 3. Apologies for the confusion; the main text is not clear on that point. Our line-search scheme is a stochastic variant of the standard Armijo line-search and our implementation does use a backtracking line-search with exponentially decaying step. Please refer to the pseudo-code in Algorithm 1 of Appendix F. We assume access to the largest step-size to simplify the theoretical analysis. The same bounds hold up to a constant factor depending on the exponential decay parameter for the backtracking line-search. We have clarified this in the main text in Section 3 and added a more detailed explanation for the backtracking line-search in Appendix A.
>
> 4. No, as long as initial step-size for the backtracking line-search is large enough, $\eta_0 \geq 2 (1 - c) / L_{\max}$, the condition ensures that all step-sizes are large enough by recursion, as $2 (1 - c) / L_{\max} \leq \eta_k \leq \eta_{k-1}$. To ensure this, in practice, we initialize $\eta_0$ to a very large value (equal to 1000 for example). However, if $\eta_0 \leq 2 (1 - c) / L_{max}$, then $\eta_k$ will remain constant and equal to $\eta_0$ since the line-search condition will be immediately satisfied and there will be no backtracking. The same reasoning holds for the stochastic Armijo line search method.
>
> 5. In Défossez et al., 2020, the authors show the convergence of AdaGrad to the solution (because the effective step-size keeps decreasing). They interpret Adam (or AMSGrad) as a constant step-size variant of AdaGrad. For these methods, the step-size is not guaranteed to decrease, and similar to us, they show convergence to a neighborhood of the solution.

---

### Author Response · Authors · 2020-11-25
**Final comments during rebuttal**

We would like to thank the reviewers for their feedback and suggestions. We have addressed the reviewers’ comments about our theoretical results and the required assumptions. We have updated the paper with further clarifications.

We would like to emphasize that the goal of our theoretical results is to understand the effect of the step-size, momentum and over-parameterization on the performance of adaptive gradient methods. Although popular, these methods are poorly understood and we need some simplifying assumptions to focus on the things that influence their empirical performance. Furthermore, the proposed line-search methods retain the theoretical convergence guarantees of adaptive gradient methods, while resulting in consistent and substantial empirical improvements over the whole range of convex and non-convex benchmarks we consider.

---

### Decision · Program_Chairs · 2021-01-07
**Final Decision**

**Decision:**

Reject

**Comment:**

Dear authors,

the paper contains many interesting and novel ideas. Indeed, tuning step-size is very time and energy-consuming, and deriving and analyzing new adaptive algorithms has not only theoretical benefits but, more importantly, is a key when training more complicated ML models.

The paper contains many weaknesses as noted by reviewers. I know that you have addressed many of them one of the reviewers is still concerned about the other issues involving Theorem 1 and the assumption of the bounded preconditioner.
He thinks the preconditioner bound is troublesome. In the overparameterized regime, he would expect the gradients to become near zero as the algorithm converges, which would actually cause the preconditioner to NOT be bounded below. It seems that the analysis might actually improve if the authors abandoned AMSGrad/Adam and instead just considered SGD for which the preconditioner assumption is not an assumption but just a property of the algorithm.





Thank you